# BINARY AUTOENCODER FOR MECHANISTIC INTERPRETABILITY OF LARGE LANGUAGE MODELS

## ABSTRACT

Existing works are dedicated to untangling atomized numerical components (*features*) from the hidden states of **L**arge **L**anguage **M**odels (LLMs) for interpreting their mechanism. However, they typically rely on autoencoders constrained by some implicit training-time regularization on **single** training instances (i.e., $L_1$ normalization, top-k function, etc.), without an explicit guarantee of **global** sparsity among instances, causing a large amount of dense (simultaneously inactive) features, harming the feature sparsity and atomization. In this paper, we propose a novel autoencoder variant that enforces minimal entropy on minibatches of hidden activations, thereby promoting feature independence and sparsity across instances. For efficient entropy calculation, we discretize the hidden activations to 1-bit via a step function and apply gradient estimation to enable backpropagation, so that we term it as **B**inary **A**uto**e**ncoder (BAE) and empirically demonstrate two major applications: **(1) Feature set entropy calculation**. Entropy can be reliably estimated on binary hidden activations, which we empirically evaluate and leverage to characterize the inference dynamics of LLMs and In-context Learning. **(2) Feature untangling**. Similar to typical methods, BAE can extract atomized features from LLM's hidden states. To robustly evaluate such feature extraction capability, we refine traditional feature-interpretation methods to avoid unreliable handling of numerical tokens, and show that BAE avoids dense features while producing the largest number of interpretable ones among baselines, which confirms the effectiveness of BAE serving as a feature extractor[1].

## 1 INTRODUCTION

Current practice for untangling atomized numerical components (*features*) from **L**arge **L**anguage **M**odels (LLMs), such as **S**parse **A**utoencoder (SAE) (Shu et al., 2025a), with training-time regularization (e.g., $L_1$ normalization on hidden activations) to implicitly atomize features **sample-wisely**. However, such methodologies do not ensure **global** sparsity, often leading to frequently activated (dense) features alongside inactive (dead) features (Stolfo et al., 2025; Rajamanoharan et al., 2024b; Sun et al., 2025), contradicting the sparsity assumption (Elhage et al., 2022) in mechanistic interpretability, as broad activations across samples hinder consistent and meaningful interpretation, and reduce the parameter efficiency by dead features.

Therefore, in this paper, to address this issue, we propose **B**inary **A**uto**e**ncoder (BAE), utilizing information-theoretic constraints among **minibatches** of training instances to address the aforementioned issue. As shown in Fig. 1, we design training objectives that minimize the entropy of hidden activation from minibatches to reduce feature covariation while enforcing global sparsity to suppress frequently activated features. However, typical hidden activations are real vectors, whose entropies are extremely difficult to calculate (Greenewald et al., 2023) as the computation requires high-dimensional numerical integrations, which involve an exponential explosion of computational complexity. To this end, we round the activations to binary, and calculate entropy on such binary vectors to significantly reduce the computational complexity of the entropy objective and utilize gradient estimation (Hubara et al., 2016) to enable the backpropagation on such a rounding operation.

We empirically demonstrated the benefits of BAE in application as: **(1) Efficient estimation of entropy from hidden state sets.** We can estimate the entropy for reconstructing the input sets as the

---

[1]The source code and the BAE checkpoints will be released upon acceptance of this paper.

hidden activation entropy with significantly reduced cost, which can serve as an important metric for understang the inner operation of LLMs. By experiment on synthetic datasets with different ground-truth entropy values, we confirm the accuracy of such entropy estimation. Moreover, we investigate the entropy dynamics inherent in the feed-forward process of standard language modeling in LLMs, revealing an information bandwidth at each layer and implicit context windows. Also, we further interpret **In-c**ontext **L**earning (ICL) (Dong et al., 2022) as a form of information reduction.

And **(2) Sparse feature extraction.** As also applied to normal SAE, the line vectors of the second autoencoder layer (sometimes termed as the *dictionary*) can serve as the atomized features extracted from the inputs. In typical SAE, features are activated densely (Stolfo et al., 2025; Rajamanoharan et al., 2024b; Sun et al., 2025) due to the inconsistent scales across hidden activation channels, rooted in the sample-wise regularization, which overlooks the global sparsity. In contrast, BAE significantly suppresses dense features as well as dead features by the entropy-based training objective on mini-batches and consistent activation scaling based on the information gain of each channel, extracting the largest number of active and interpretable features. Meanwhile, we refine the traditional feature interpretation method (Bills et al., 2023; Huben et al., 2024), thereby avoiding unreliable handling of numerical tokens by LLMs (Press et al., 2023), for more robust automatic feature interpretation.

**Main findings and contributions:** (1, §3) We propose **B**inary **A**uto**e**ncoder (BAE), a novel variant of autoencoder with binarized hidden activation and training-time entropy constraint. (2, §4) BAE enables accurate entropy calculation on hidden state sets, providing an important tool for analyzing LLM behavior from an information perspective. With this, we observe LLM inference as an information increment: hidden states at each layer act as channels with increasing bandwidth w.r.t. the layer depth, where each token adds information until the limit is reached, while ICL appears as information reduction. (3, §5) We confirm BAE as an effective atomized feature extractor with no dense features and the largest amount of interpretable features extracted. (4, §5) We revise the typical automatic feature interpretation (Bills et al., 2023; Huben et al., 2024) methods working on the line vector of the second BAE/SAE layer, which forces LLMs to process numerical tokens unreliably (Press et al., 2023), for interpreting the features extracted by BAE more robustly.

## 2    RELATED WORKS

**Dictionary Learning** (Shu et al., 2025b). Modern mechanistic interpretability (Sharkey et al., 2025; Bereska & Gavves, 2024) views LLM hidden states as superpositions of atomic features, motivating efforts to disentangle them for better semantical understanding of the LLMs' operation. Originally, **S**parse **A**uto**e**ncoder (SAE), an autoencoder with training-time $L_1$ normalization on the hidden activations for the sample-wise sparsity, is utilized for such disentangling (Bricken et al., 2023; Huben et al., 2024; Templeton et al., 2024; Gao et al., 2025). Additionally, variants of SAE are proposed to enhance the performance by utilizing modified activation functions (Gao et al., 2025; Bussmann et al., 2024; Rajamanoharan et al., 2024a;b) or merging multiple-layer hidden states from LLMs (Shi et al., 2025; Lindsey et al., 2024). However, these methods do not warrant a global sparse decomposition of hidden states, that is, with some guarantee to minimize the activation frequency as well as covariation among channels of hidden activations to reconstruct the original hidden states with sparse atomized features. Such a drawback causes the activations to be dense (Kissane et al., 2024; Rajamanoharan et al., 2024b; Sun et al., 2025; Stolfo et al., 2025), i.e., some features are activated over diverse inputs lacking a common semantic theme, and thus cannot be uniquely and clearly attributed (or interpreted) semantically. Therefore, these dense features can be regarded as a merger of multiple features, i.e., insufficient atomization, which is the main motivation of this paper: explicitly promote the global feature sparsity and atomization by improved training methods, to capture a larger amount of interpretable features, and to improve the accuracy of feature interpretation.

**Discrete-valued Neural Networks.** To utilize the information-theoretic constraints mentioned before, we round the hidden activations to binary. The framework of such binary neural networks, including rounding the real activation into 1-bit, and estimating the gradient for such rounding operation, etc., is originally proposed by Hubara et al. (2016). Subsequently, numerous variants are proposed primarily based on different rounding operations and gradient estimations (Rastegari et al., 2016; Zhou et al., 2016; Choi et al., 2018; Vargas et al., 2024). A detailed survey can be found in Qin et al. (2020). In this paper, we utilize these methodologies for the autoencoder with binarized hidden activations to reduce the calculation cost on the information-theoretic constraints.

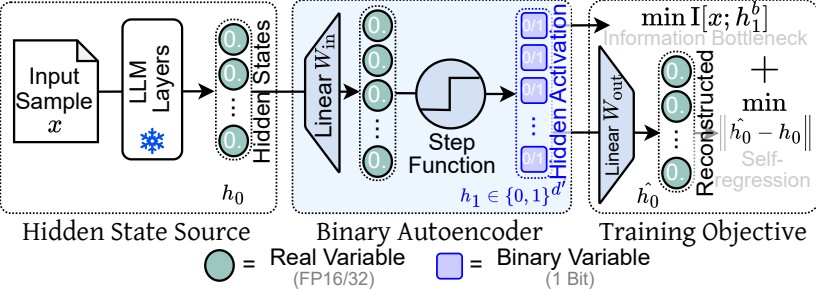

Figure 1: Feed-forward computation and training objective of BAE. Hidden states $h_0$ from LLM layers are mapped by $W_{\text{in}}$, binarized into $h_1$ via a step function, and projected back by $W_{\text{out}}$ as $\hat{h_0}$. The $\hat{h_0}$ is fed to the self-regression loss, while $h_1$ is fed to the information bottleneck loss.

## 3 BINARY AUTOENCODER

To address SAE's drawbacks, we propose **B**inary **A**uto**e**ncoder (BAE) shown in Fig. 1 and follows:

**Feed-forward Calculation.** Given input hidden state $h_0 \in \mathbb{R}^d$ from an LLM, a binary autoencoder $\mathcal{F}$ conducts such calculation for the output $\mathcal{F}(h_0)$ (also noted as $\hat{h_0} = \mathcal{F}(h_0)$):

$$\mathcal{F}(h_0) = \Gamma\left(h_0 W_{\text{in}}\right) W_{\text{out}} + b, \tag{1}$$

where $W_{\text{in}} \in \mathbb{R}^{d \times d'}$, preforming a linear decomposition on input $h_0$ to $d'$ dimensions, $b \in \mathbb{R}^d$ is a bias term to reconstruct the anisotropism (Gao et al., 2019; Ethayarajh, 2019; Biś et al., 2021; Godey et al., 2024) of the hidden states which contains minor information thus should be ignored as background value, $\Gamma$ is the quantization function, projecting $\mathbb{R}^{d'}$ into $\{0,1\}^{d'}$ element-wisely. Here, we use the step function to binarize the $h_0 W_{\text{in}}$ into hidden activation $h_1$:

$$\Gamma\left([x_1, x_2, \ldots, x_{d'}]\right) = [\gamma(x_1), \gamma(x_2), \ldots, \gamma(x_{d'})],$$
$$\gamma(x) = \begin{cases} 0, x < 0 \\ 1, x \geqslant 0. \end{cases} \tag{2}$$

The $\Gamma$ also provides essential non-linearity for the numerical expressivity of Eq. 1 (otherwise, the $W_{\text{in}} W_{\text{out}}$ will degrade into one matrix).

**Self-regression Training Objective.** Given a minibatch of hidden states (with batch size $n_b$) $H_0 = \left\{h_0^{(1)}, h_0^{(2)}, \ldots, h_0^{(n_b)}\right\}$, we calculate the self-regression training objective with $L_2$ norm:

$$\mathcal{L}_r(H_0) = \frac{1}{n_b} \sum_{h_0 \in H_0} \|h_0 - \mathcal{F}(h_0)\|_2. \tag{3}$$

**Information Bottleneck Constraint.** To constrain the hidden activation ($h_1 = \Gamma(h_0 W_{\text{in}})$) to a global sparse representation for $h_0$, we minimize the margin entropy of $h_1$ (notice that, since $h_1 \in \{0,1\}^{d'}$, such calculation are differentiable and highly efficient, without any numerical integration on the real space). Also, to maximize the effectiveness of the constraint on margin entropy, we also penalize the covariance (except the diagonal elements) of $h_1$ to force the margin entropy to approach the joint distribution entropy. That is, on the minibatch $H_0$, we define the entropy-based loss term:

$$\mathcal{L}_e(H_0) = \alpha_e \mathrm{H}[\frac{1}{n_b} \sum_{h_0 \in H_0} \Gamma(h_0 W_{\text{in}})] + \alpha_c \mathrm{D}[\Gamma(H_0 W_{\text{in}})],$$

$$\text{where } \mathrm{H}[x] = -\sum_{i=1}^{d'} x_i \log_2 x_i, \ \mathrm{D}[X] = \sum_{i,j:i \neq j} |\text{cov}(X)_{i,j}|. \tag{4}$$

The $\alpha_e$ and $\alpha_c$ are hyperparameters. The total loss can be written as:

$$\mathcal{L}(H_0) = \mathcal{L}_r(H_0) + \mathcal{L}_e(H_0). \tag{5}$$

This training objective resembles and similates the information bottleneck methods (Tishby et al., 2000; Kawaguchi et al., 2023; Tishby & Zaslavsky, 2015), where $\mathcal{L}_e$ minimizes the mutual information between input $h_0$ and latent $h_1$, while $\mathcal{L}_r$ maximizes that between output and $h_1$.

**Gradient Estimation for $\Gamma$.** Since the differential of binarization function $\Gamma$ is 0 almost everywhere, to enable the backpropagation from $\mathcal{L}$ to $W_{\text{in}}$, following the previous works (Hubara et al., 2016; Vargas et al., 2024), we estimate the gradient of $\Gamma$ by smoothing function $x \mapsto (1 + e^{-x})^{-1}$ ("sigmoid") elementwisely. Therefore, the differential of $\Gamma$ is estimated as:

$$\frac{\partial \Gamma(x)}{\partial x} := x \odot (\mathbf{1} - x), \qquad (6)$$

where $\mathbf{1}$ is all-ones vector, $\odot$ is the Hadamard product of two vectors.

**Default Hyperparameters.** We defaultly set $d' = 4d$ (i.e., the expanding rate = 4), the entropy loss weight[2] $\alpha_e = 10^{-7}$, the covarience loss weight $\alpha_c = 10^{-7}$. We use Adam (Kingma & Ba, 2014) optimizer with learning rate $5 \times 10^{-4}$, momentum factor $\alpha_1 = 0.9, \alpha_2 = 0.999$, and minibatch size $n_b = 512$ for 2000 epochs, with $\alpha_e = 0$ in the first 500 epochs.

## 4 ENTROPY ESTIMATION OF HIDDEN STATES BY BAE

**Entropy Estimation by BAE.** As a direct measurement of information amount, calculating entropy for hidden states in neural networks can promote a closer observation into the inner mechanism. However, directly computing the differential entropy of a high-dimensional vector set requires probability **d**ensity **e**stimation (DE) and integration, which is neither accurate (DE suffers from the curse of dimensionality and floating-point errors) nor efficient (numerical integration costs $\mathrm{O}(C^d)$ for $d$ dimensions with $C$ integration cells) (Greenewald et al., 2023). In contrast, BAE can improve such entropy calculation by best-effortly decomposing the original real-value vectors into binary vectors ($h_1$) with pair-wisely independent elements, enabling efficient entropy estimation by margin entropy of the mean hidden activation $\bar{h}_1$: given a vector set $H_0 = \left\{ h_0^{(i)} \right\}_{i=1}^{n}$, we

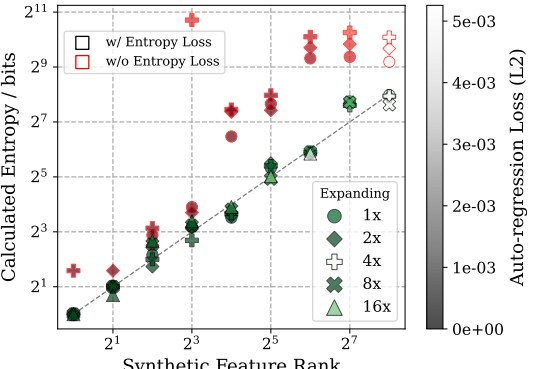

Figure 2: Evaluation of BAE entropy calculation on the synthetic dataset. Horizontal axis: rank $r$ of the current dataset, vertical axis: calculated entropy, the green/red color refers to whether $\mathcal{L}_e$ is enabled, and the opacity refers to the $\mathcal{L}_r$ on the whole input set.

encode them into $h_1$s by a trained BAE as $H_1 = \left\{ h_1^{(i)} = \Gamma \left( h_0^{(i)} W_{\text{in}} \right) \right\}_{i=1}^{n}$. Since the $h_1$s are binarized and best-effortly decorrelated pairwise by the covariance loss, we can calculate the *entropy required for reconstructing $H_0$* as $\mathrm{H}[\bar{h}_1]$, where $\bar{h}_1$ are the averages among the line vectors of $H_1$.

In this section, we evaluate the aforementioned entropy estimation on a synthesis dataset to show its accuracy, and utilize such entropy calculation to track the feed-forward process of LLM.

### 4.1 EVALUATING ENTROPY CALCULATION FROM BINARY AUTOENCODER

**Synthetic Directional Benchmarking.** To evaluate the entropy estimation of BAE's hidden binary activations ($h_1$), we build a synthetic random directional benchmark: **(1)** sample a $d$-dimensional $r$-rank orthonormal basis $M \in \mathbb{R}^{r \times d}$, **(2)** sample $r$ binary coefficients $c \in {0, 1}^r$, and **(3)** generate an instance $cM$ by summing selected $M$ basis from 1-elements in $c$. Repeat (2) and (3) $n$ times, we get a synthesis random directional dataset with $n$ samples. Intuitively, the entropy of the dataset is $r$ since the only randomness comes from the $r$ independent Bernoulli coefficients given the fixed basis. We train the BAE on a set of such datasets with various $r$ (details in Appendix A.1), and confirm whether the entropies calculated following the aforementioned method hit the corresponding $r$.

**BAE can Accurately Estimate Entropy of Synthetic Vector Set.** The evaluation results are shown in Fig. 2, where the standard BAE implementation with normal entropy objective (green)

---

[2]Notice that, considering the magnitude around the late training stage (as shown in Fig. 2 and 3) of the self-regression loss ($\mathcal{L}_r$, around $10^{-3} \sim 10^{-2}$) and the margin entropy ($\mathrm{H}[\cdot]$, around $10^0 \sim 10^3$), although the weights $\alpha_e$ are set quite small, the magnitude of the $\alpha_e \mathrm{H}[\cdot]$ are balanced as a regularization term against $\mathcal{L}_r$.

hint the corresponding $r$ accurately, despite of the inner dimension $d'$ (noted as the expending rate $d'/d$, shown as the marker shape in Fig. 2). In contrast, the adversarial experiments (red), where the $\alpha_e$ and $\alpha_c$ are both set to 0, show clearly higher calculated entropy, suggesting that BAE with entropy constraint can find the essential (minimal) entropy values to reconstruct the input sets.

**Natural Low-entropy Tendency.** Moreover, interestingly, when the estimated entropy constraint is disabled (red in Fig. 2), the entropy gathers higher but near the diagonal, suggesting that BAE without entropy training objective can find a relatively low-entropy encoding $h_1$, even if no normalizations are conducted to minimize the encoding entropy. To get a closer observation, we plot the self-regression loss $\mathcal{L}_r$ and training-time entropy $\mathcal{L}_e$ on ev-

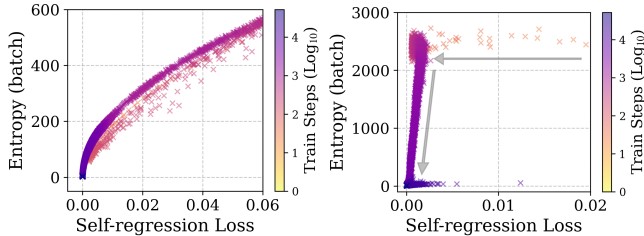

Figure 3: Training dynamics of BAE (**left**) with entropy objective and (**right**) without entropy objective. The horizontal axes are the self-regression training loss ($\mathcal{L}_r$), and the vertical axes are the entropy calculated from the training batch.

ery training step, with $r = 2$ (more cases in Appendix D.2), as shown in Fig. 3, with normal setting (left) or $\alpha_e = \alpha_c = 0$ (right). In the normal setting (left), a stable and monotonous convergence to both minimal entropy and self-regression loss can be observed. However, if the entropy loss is disabled (right), the training dynamics become multiphase and non-monotonic. Specifically, **Phase 1:** the loss rapidly decreases while the hidden activations maintain relatively high entropy; **Phase 2:** entropy sharply drops to near zero, with the loss remaining low, even if no entropy and weight penalty are utilized; and **Phase 3:** both loss and entropy oscillate in a narrow range at the end of training. Such an observation suggests that: Gradient descent on simple regression loss finds relatively "simple" representations, after a long-term stagnation on loss value, supporting previous work on training dynamics (Tishby & Zaslavsky, 2015; Saxe et al., 2018; Huh et al., 2023; Nanda et al., 2023; Shah et al., 2020; Frankle & Carbin, 2019; Bartlett et al., 2020). Additionally, enabling the entropy penalty suppresses the harmful Phase 3 oscillations, confirming its effectiveness.

## 4.2 Tracing the Normal Sentence Modeling by Entropy

In this section, we track the LLM's feed-forward dynamics by the entropy of hidden states. In detail, we sample $n = 262144$ sentences from Pile (Gao et al., 2020), inputting them into Llama 3.2-1B[3] (Grattafiori et al., 2024), then extract the hidden states of $2^{0,1,\cdots,10}$-th tokens from each layer. Then we train a BAE on each layer and position, with experiment details in Appendix A.2. From the trained BAE, we calculate the entropy as shown in Fig. 4, where we observe:

**Layer Bandwidth.** The entropy of hidden states from a specific layer increases with the number of prefix tokens and eventually saturates at a fixed value. This observation suggests that: if the hidden space of a specific layer is viewed as a channel (Elhage et al., 2021) through which token information is communicated, then the channel has a fixed bandwidth, limiting the amount of accommodated token information. That is, Transformers have implicit context windows on each layer, where the exceeded information is discarded or distorted. Unlike convolutional neural networks, we have no evidence to infer that such context windows are contiguous or even binary, so that it is possible to overwrite parts of but not the whole information from an old token by new information, causing distortion like the *lost-in-the-middle* problem (Hsieh et al., 2024; Liu et al., 2024a; He et al., 2024; An et al., 2024; Liu et al., 2025).

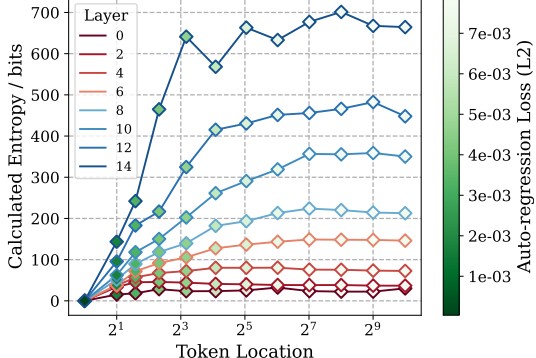

Figure 4: Entropy calculated on the hidden states extracted from specific layers and token locations from Pile and Llama 3.2-1B. The curve colors refer to the extracted layers, the scatter colors refer to the $\mathcal{L}_r$ on the whole input set.

---

[3]Datasets and models in this paper are loaded from HuggingFace.

**Token Information Gain.** As shown in Fig. 4, for a specific token position, hidden states from a deeper layer carry more information, and the entropy also saturates later on a deeper layer, suggesting that Transformer blocks sequentially inject contextualized information into hidden states. As a result, deeper layers are more likely to simulate broader context windows, which facilitates the processing of downstream tasks at higher levels of information aggregation and abstraction. In contrast, shallower layers, restricted by narrower context windows, tend to focus on local linguistic-level features and propagate this information to the later layer to combine into broader-scoped abstraction, which is consistent with intuition and previous empirical observations (Jawahar et al., 2019; Chen et al., 2023; Wang et al., 2023; Xiao et al., 2025). We discuss this point deeper in §6.

### 4.3 TRACING IN-CONTEXT LEARNING BY ENTROPY

In this section, we track the LLM's ICL (refer Appendix A.3 for introduction) inference dynamics via entropy, similar to §4.2. In detail, we sample $262144$ ICL input instances from SST-2 (Socher et al., 2013) with specific demonstration numbers, then extract the hidden states of the last token (the ":" in the "sentiment:", where the answer to the query will be generated) of a specific layer, and then train a BAE on these hidden states, with prompt templates, settings and parameters detailed in Appendix A.3. We calculate the entropy as shown in Fig. 5, where: **(1)** similar to the normal language modeling, a deeper layer contains more information, which suggests that the implicit context windows hypothesis still stands for the ICL scenario. However, **(2)** contradicts the nor-

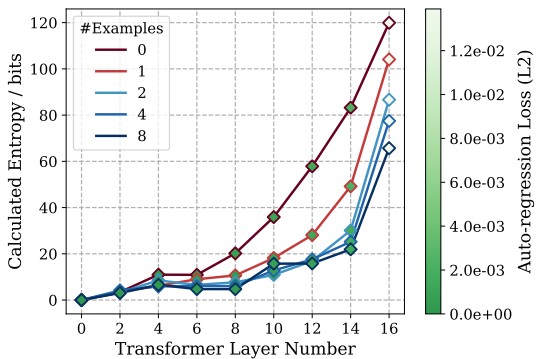

Figure 5: Entropy calculated on the hidden states extracted from specific layers and the last token from ICL inputs from SST-2. The curve colors refer to the number of demonstrations.

mal language modeling, hidden states with more demonstrations (globally longer sentence lengths and better accuracy, see Appendix A.3) counterintuitively contain lower entropy, which suggests an interesting conclusion: *ICL is achieved through removal of information*, where the information useless to the specified task may be suppressed by the given demonstrations. This contradicts the mainstream idea that ICL "learns new tasks or knowledge" (Pan et al., 2023; Li et al., 2024; Wang et al., 2025), and thus may offer a new perspective for interpreting the ICL inference processing as "deleting unrelated information from the query encoding (Cho et al., 2025) on the last token".

## 5 ATOMIZED FEATURES EXTRACTION BY BAE

Given that our entropy constraint promotes sparsity and decorrelation of hidden activations $h_1$, the linear decomposition from $h_0$ to $h_1$ can disentangle atomized features in LLM hidden states (Elhage et al., 2022; Hänni et al., 2024). To evaluate such disentanglement, in this section, we assign human-interpretable semantics to each channel of $h_1$ and assess their consistency across inputs, using an improved automatic interpretation method under the LLM-as-a-judge framework (Gu et al., 2024).

### 5.1 COMMON SEMANTICS-BASED FEATURE INTERPRETATION AND EVALUATION

**Revisiting Current Automatic Feature Interpretation and Evaluation.** We begin with a revisit to the current automatic interpretation and evaluation methods of features (Bills et al., 2023; Huben et al., 2024), where for a channel in $h_1$ corresponding to a feature (line vector) in $W_{out}$, **(Step 1)** given one input text, the activation magnitudes (e.g., the value of the specified channel in $h_1$ of SAE) on the channel are calculated on every token, then **(Step 2)** the tuple of sentence and activation magnitudes on all tokens is input in LLMs with a prompt (e.g., "Please predict the explanation of the feature given the following activations.") for interpreting such feature into one phrase (e.g., "freedom-related terms"). Then, **(Step 3)** given a test input text, a simulation of the activation magnitudes based on the generated interpretation for each token is queried from the LLM, and the correlation coefficient of the simulated activation magnitudes and the SAE-calculated activation magnitudes is regarded as the interpretability score of this feature (refer to Bills et al. (2023) for details).

Notice that Step 2 of the above process relies on the LLM's ability to accurately handle large amounts of numerical tokens to generate reliable explanatory phrases, and Step 3 requires the LLM to faithfully simulate activation magnitudes by numerical tokens, which places a high demand on the LLM's mathematical reasoning capabilities and also output calibrations. However, current research indicates that LLMs exhibit weaker capabilities in numerical reasoning compared to linguistic tasks (Press et al., 2023; Schick et al., 2023; Ahn et al., 2024; Xu et al., 2025). Also, the output can be implicitly biased to some specific tokens (Zhao et al., 2021; Geng et al., 2024), making the aforementioned pipeline unreliable, requiring a revision to improve the robustness and credibility.

**Common Semantics-based Feature Interpretation and Evaluation (ComSem).** Therefore, to avoid relying on LLMs to directly process numerical tokens, we propose ComSem as a new pipeline that leverages the LLMs' strength in linguistic semantic recognition to interpret the extracted features. In detail (refer Appendix A.4 for the detailed pseudocode): for a specific channel (corresponding to a feature) in the $h_1$ of an autoencoder, given a set of test sentences, **(Step 1)** for all the tokens in the set, whose hidden states have *significant activation magnitude* on the specific channel (detailed in §5.2 for BAE), we collect them along with their sentences. **(Step 2)** We query the backend LLM to find the commonality of these tokens presented in their sentences, as the interpretation of the feature corresponding to the channel. **(Step 3)** On a hold-out set of such token-sentence tuples significantly activated on the channel, we query the LLM to judge whether these tokens can be interpreted by the generated interpretation from Step 2. The ratio of the "Yes" answer is calculated as the interpretability score of this feature. ComSem can avoid the concern in mathematical and numerical reasoning, and provides the possibility of applying simple output calibrations[4] on the true-or-false output. So, we utilize ComSem to evaluate the BAE, as described below.

## 5.2 Extracting, Interpreting and Evaluating BAE Features by ComSem

To utilize the ComSem pipeline on BAE, we first propose a rescaling method to identify the *significant activations* (utilized in Step 1 of ComSem) from BAE, since BAE's binary hidden activations cannot directly reflect the activation magnitudes. Then, we evaluate the interpretability score of BAE-extracted features by ComSem to demonstrate that BAE is an effective feature extractor.

**Estimate the Activation Magnitude from the Feature Burstiness.** Since our aforementioned ComSem pipeline requires access to feature activation *magnitude*, which our BAE can not explicitly provide[5], as shown in Fig. 10, we calculate the *burstiness* of each channel to convert the binary activations $h_1$ into the activation magnitudes. In detail, for each (indexed by $i$) instance $h_1^{(i)}$ in a hidden activation set $H_1 = \{h_1^{(i)}\}_{i=1}^n$, we compare the distance of $h_1^{(i)}$ with a prior distribution[6] $\bar{h}_1$ as $\beta^{(i)} = \log_2 |h_1^{(i)} - \bar{h}_1|$, where the $\log_2$ is the element-wise logarithm. Such $\beta^{(i)}$ is the channel-wise activation magnitude of $h_1^{(i)}$, where a channel $j$ with larger $\beta_j^{(i)}$ suggests the $j$-th feature from $h_1^{(i)}$ is more bursty. A more bursty feature reduces more uncertainty to reconstruct the $h_0$ so that *carries more information against the prior distribution $\bar{h}_1$, which makes it more representative of* $h_0$. Then, for each instance, we simply select the indices with the top-$k$ greatest values in $\beta^{(i)}$ as the significant activation channels utilized in Step 1 of ComSem.

**Such burstiness calculation has the following advantages: (1) Global sparsity induced by low entropy.** Notice that our entropy training objective shown in Eq. 4 is actually punishing the burstiness summed from all the input samples on all the channels among the minibatches, therefore, the calculated $\beta^{(i)}$ for each sample is numerically sparse in two directions, i.e., **(i)** for one instance $h_1^{(i)}$, most of elements in $\beta^{(i)}$ are suppressed to be sufficient small values, and **(ii)** for one feature channel $j$, most of $\beta_j^{(i)}$ are small values among $i$ among all instances, which is aligned to the sparsity assumption of mechanistic interpretability (refer to Appendix D.1 for visualization).

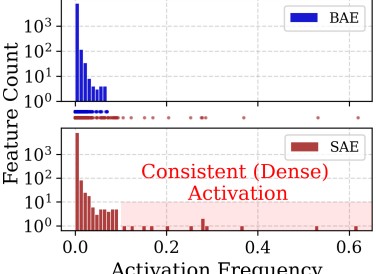

Figure 6: Feature activation frequency distribution of Layer 11 (more layers in Appendix D.3).

---

[4]E.g., described in (Zhao et al., 2021; Fei et al., 2023). Such output calibration is not utilized in this work.

[5]Note that, in typical SAE, the hidden activation (congener of our $h_1$) can directly serve as such magnitude.

[6]Such $\bar{h}_1$ can either be an accompanying component saved during training, or mean value on $H_1$ set.

**(2) Consistent scale and clear interpretability.** In typical SAEs, different channels of $h_1$ lie on varying scales, making some inherently high-magnitude channels appear consistently active ("dense features", Appendix D.1 and Fig. 6) (Stolfo et al., 2025; Rajamanoharan et al., 2024b) while others seem inactive ("dead features"). This issue is particularly pronounced when using top-$k$ selection based on absolute activation values to identify significant features, reducing the number and introducing confusion to activated features. Therefore, intuitively, as shown in §5.3, rescaling activations into an aligned distribution alleviates this problem in typical SAE. And in our pipeline, BAE simplifies the rescaling since each channel follows a Bernoulli distribution and requires only one statistic ($\bar{h_1}$), whereas the real-valued $h_1$ in typical SAEs is harder to characterize and rescale.

## 5.3 BAE IS AN EFFECTIVE INTERPRETABLE FEATURE EXTRACTOR

As mentioned before, BAE effectively mitigates the two issues in feature extraction of typical SAE: dense features on inherently high-magnitude channels, which lower consistency among instances with the same feature activated (i.e., low interpretability scores in Step 3 of Com-Sem), and dead features on inherently low-magnitude channels, leaving many channels idle. In this section, we empirically confirm the mitigation of both issues by BAE with ComSem on BookCorpus (Zhu et al., 2015) against SAE variants trained on Pile (detailed in Appendix A.4) as shown in Table 1, where compared to all the baselines, BAE can extract the largest amount of active features from the LLM's hidden states, with a considerable interpretability score. However, one can doubt that the interpretability score of BAE is "not SotA" in the Table 1. Our explanation is: BAE has a strong feature extracting ability, so that it captures some complex or vague features (refer to Appendix C) which are difficult to interpret by natural language from LLMs, which reduces the averaged interpretability scores. As shown in Appendix B, if some of the low-score features are removed, then BAE reaches the SotA of interpretability score and also remains the largest feature amounts.

Table 1: Evaluation of BAE and baselines on 2 backend LLMs ($d'/d = 4$). **Feature Activated**: The number of features where sufficient $h_0$ instances (more than 8, refer to Appendix A.4) with significant activation magnitude are observed on the corresponding channel. **FI#**: The number of feature channels with non-zero Com-Sem interpretability score. **Score**: The averaged Com-Sem score among all activated channels.

| Feat. Source | Model (Appendix A.4) | Feature Activated | ComSem$_{4.1\text{-mini}}$ FI# | ComSem$_{4.1\text{-mini}}$ Score | ComSem$_{4.1}$ FI# | ComSem$_{4.1}$ Score |
|---|---|---|---|---|---|---|
| Llama 3.2-1B Layer 5 $d = 2048$ | ReLU SAE | 1922 | 1103 | 0.216 | 1251 | 0.294 |
| | Top-k SAE | **3531** | **2614** | **0.439** | 2763 | **0.445** |
| | Gated ReLU SAE | 17 | 11 | 0.382 | 6 | 0.294 |
| | ReLU SAE Resc. | 567 | 309 | 0.233 | 367 | 0.269 |
| | TransCoder | 335 | 218 | 0.275 | 195 | 0.238 |
| | **BAE (ours)** | 3012 | 2249 | 0.422 | 2449 | 0.439 |
| Llama 3.2-1B Layer 9 $d = 2048$ | ReLU SAE | 1914 | 1128 | 0.229 | 1274 | 0.254 |
| | Top-k SAE | 3179 | 2389 | 0.450 | 2542 | 0.461 |
| | Gated ReLU SAE | 100 | 85 | **0.638** | 80 | **0.596** |
| | ReLU SAE Resc. | 3161 | 2028 | 0.258 | 2190 | 0.294 |
| | TransCoder | 417 | 257 | 0.260 | 263 | 0.264 |
| | **BAE (ours)** | **4675** | **3370** | 0.370 | **3624** | 0.394 |
| Llama 3.2-1B Layer 11 $d = 2048$ | ReLU SAE | 2065 | 1177 | 0.232 | 1380 | 0.260 |
| | Top-k SAE | 3417 | 2540 | 0.440 | 2684 | 0.452 |
| | Gated ReLU SAE | 1226 | 976 | **0.531** | 1026 | **0.557** |
| | ReLU SAE Resc. | 744 | 435 | 0.259 | 482 | 0.278 |
| | TransCoder | 1794 | 979 | 0.218 | 1090 | 0.239 |
| | **BAE (ours)** | **5464** | **3882** | 0.360 | **4140** | 0.382 |
| Llama 3.2-1B Layer 14 $d = 2048$ | ReLU SAE | 2528 | 1423 | 0.195 | 1600 | 0.217 |
| | Top-k SAE | 2702 | 1900 | 0.389 | 2004 | 0.418 |
| | Gated ReLU SAE | 2948 | 2095 | **0.412** | 2250 | **0.435** |
| | ReLU SAE Resc. | 3962 | 2361 | 0.252 | 2661 | 0.274 |
| | TransCoder | 3401 | 1931 | 0.237 | 2166 | 0.267 |
| | **BAE (ours)** | **6120** | **3963** | 0.324 | **3971** | 0.323 |
| Llama 3.2-3B Layer 20 $d = 3072$ | ReLU SAE | 1923 | 1183 | 0.289 | 1289 | 0.312 |
| | Top-k SAE | 3234 | 2286 | 0.402 | 2508 | 0.425 |
| | Gated ReLU SAE | 4628 | 3271 | **0.402** | 3580 | **0.437** |
| | ReLU SAE Resc. | 2122 | 1222 | 0.250 | 1451 | 0.294 |
| | TransCoder | 5508 | 3001 | 0.233 | 3424 | 0.257 |
| | **BAE (ours)** | **9308** | **5956** | 0.308 | **6805** | 0.348 |

Moreover, we conduct normalization[7] to the $h_1$ of ReLU SAE for a rescaled activation magnitude, and evaluate as "ReLU SAE Resc." in Table 1, where comparing to the vanilla ones, the interpretability scores are observably improved by the normalization rescaling. Since the training of SAE ignores cross-sample sparsity and its $h_1$ distribution is hard to estimate, the effectiveness of rescaling remains limited.

**Activated Feature Distribution.** To examine BAE's effective suppression towards dense feature, we visualize the activation frequency distribution of each channel on the trained BAEs and SAEs as Fig. 6. In the visualizations, BAE features are sparsely activated with a left-leaning distribution, while some of the feature channels in the typical SAE keep high activation frequencies with a long-tail distribution, suggesting dense activations (Stolfo et al., 2025; Rajamanoharan et al., 2024b). Such results support our hypothesis that the minibatch-oriented entropy objective can mitigate the global density among input instances. Visualizations on more settings are provided in Appendix D.1.

---

[7]I.e., linearly scale every channel value to a distribution over $h_1$ instances with mean value 0 and standard deviation 1, to mitigate the influence of inherent activation magnitude of channels.

## 6 CONCLUSION AND DISCUSSION

**Conclusions.** In this paper, we propose Binary Autoencoder as a toolkit for mechanistic interpretability research for LLMs. BAE utilizes entropy training objective on a minibatch of binary hidden activations to extract globally sparse and atomized features. Our findings demonstrate that BAE can accurately estimate the entropy of a feature set and effectively disentangle atomized and sparse features from LLM's hidden states, making it a powerful tool for interpretability research.

**Compress Hidden States into Fewer Bits by BAE.** As detailed in Appendix A.5, one can utilize BAE to compress vectors by storing only channel indices of $h_1$ where the burstiness against prior $\bar{h}_1$ exceeds a threshold. Such compression has an expected coding length for each instance[8] $\text{L}_{\texttt{int}}\text{H}[h_0]$, outdistances the original $\text{L}_{\texttt{float}}d$. Reconstruction is done by flipping $\text{round}(\bar{h}_1)$ at the stored indices and projecting through $W_{\text{out}}$. We evaluate this compression with a threshold $\log 0.5$ (refer Appendix A.5 for details), as shown in Table 2, achieving compression rates as low as 1% with low MSE, indicating both effective compression and reconstruction quality. However, the reconstruction remains distorted in the radial direction, manifesting as low cosine similarity with the originals, likely arising from the $L_2$ loss of BAE, which ignores radial information. Since this method constitutes lossy compression and falls outside the paper's main focus, we do not present it as a core contribution but a potential application. For reliable and efficient storage or transmission, specialized objectives beyond simple regression are needed to preserve essential information better.

**Hypothesis of Atomic Features.** Since BAE represents feature activations with binary values rather than continuous ones, inherently continuous attributes (such as RGB values (Engels et al., 2025; Modell et al., 2025)) may be discretized into multiple features, which is reasonable since these features cannot be represented with 1-bit, so decomposing them into multiple features facilitates the correct estimation of the entropy. On the other hand, due to the narrow radial distribution of Transformer hidden states (Gao et al., 2019; Ethayarajh, 2019), different features may share the same direction but differ in

Table 2: Compression and reconstruction on Llama 3.2-1B. **Memory**: storage cost before/after compression. **MSE / Cos.Sim.**: mean square error or cosine similarity between the source and reconstruction.

| Layer# | $\alpha_e, \alpha_c$ | Memory (MB, +Model) | MSE | Cos.Sim. |
|---|---|---|---|---|
| 5 | 1e-7 | $16713 \rightarrow 141$ | 0.016 | 0.681 |
| 9 | 1e-7 | $16713 \rightarrow 173$ | 0.040 | 0.760 |
| 11 | 1e-7 | $16713 \rightarrow 166$ | 0.049 | 0.800 |
| 14 | 1e-7 | $16713 \rightarrow 178$ | 0.092 | 0.816 |
| 5 | 1e-9 | $16713 \rightarrow 170$ | 0.004 | 0.748 |
| 9 | 1e-9 | $16713 \rightarrow 183$ | 0.052 | 0.809 |
| 11 | 1e-9 | $16713 \rightarrow 206$ | 0.099 | 0.831 |
| 14 | 1e-9 | $16713 \rightarrow 270$ | 0.030 | 0.860 |

distance, which is difficult to distinguish in continuous SAEs, whereas BAE avoids this issue by discretizing them into atomic units. Also, refer to Table 1, compared to baselines, the number of discretized features extracted by BAE significantly increase against layers, with SAE features remain constant, suggesting that continuous features are more likely to emerge in later layers, which are split into more discrete features by BAE, consistent with previous works (Jawahar et al., 2019; Chen et al., 2023; Allen-Zhu & Li, 2023; Liu et al., 2024b), still needs further exploration.

**Limitations and Open Questions.** (1) **Non-zero Reconstruction Loss.** In our experiments, the regression loss remains notable, which may cause a performance degradation. Future work can be devoted to modifying the structure of BAE, especially by more advanced binarization functions (Wang et al., 2018; Vargas et al., 2024) and gradient estimation methods (Darabi et al., 2019; Yang et al., 2019); also, more scaled training sets for BAE can be utilized in an industrial scenario. (2) **Natural Language Feature Interpretation Based on Tokens.** In this paper, ComSem, and also the traditional automatic feature interpretation methods, are all based on the natural language interpretation of tokens where the features originate. However, the contextualization induced by Transformer layers may cause the semantics of the hidden states to deviate from the original token, but the method family overlooks such an effect. Future works can be devoted to directly decoding the semantics from the extracted atomized features by tools such as LogitLens (Joseph Bloom, 2024) or PatchScopes (Ghandeharioun et al., 2024). Also, not all the features can be interpreted by natural languages, such as Task Vectors (Hendel et al., 2023; Kharlapenko et al., 2024). We discuss this point in Appendix C. (3) **Investigation Scope.** Due to computational limits, we evaluate BAE only on Llama 3.2-1B and 3B, leaving large-scale tests for future work. Nevertheless, we believe this paper provides a sufficient prototype of BAE for addressing dense/dead feature issues in SAE.

---

[8]Here, $\text{L}_{\texttt{int}}$ or $\text{L}_{\texttt{float}}$ denotes the number of bits required to encode a single index for the compressed coding or one float variable for an original feature element, and $\text{H}[h_0]$ is the entropy estimated by the BAE from the distribution of $h_0$, $d$ is the dimensionaility of $h_0$. The $\text{L}_{\texttt{int}}$ is usually less than $\text{L}_{\texttt{float}}$, and $\text{H}[h_0]$ is also usually less than $d$ (as demonstrated in Fig. 4, with $d = 2048$ and $\text{H}[h_0] < 700$).

THE USE OF LARGE LANGUAGE MODELS

In this paper, LLMs are used to polish writing and also help the experiment as described in §5 under the LLM-as-a-judge framework (Gu et al., 2024).

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

# Appendices

## A  EXPERIMENT DETAILS

### A.1  SYNTHETIC DIRECTIONAL BENCHMARKING (§4.1)

In the experiment of §4.1, we utilize the hyperparameters as: original vector dimensionaility $d = 2048$, rank $r \in \{0, 1, 2, 4, 8, 16, 32, 64, 128, 256, 512\}$, $\alpha_e = 5 \times 10^{-7}$, $\alpha_c = 10^{-6}$.

We generate all $65536$ samples, with $n = 52428$ training samples and the remaining as validation samples, with non-mentioned hyperparameters kept as the default. After the training, we run BAE again on all $65536$ samples before for binary encoding $h_1$, and calculate the marginal entropy on these $h_1$ as mentioned in Eq. 4.

### A.2  TRACING NORMAL SENTENCE MODELING BY ENTROPY (§4.2)

In the experiment of §4.2, we generate $n = 209715$ data samples for the training of BAE, with more $52429$ data samples as the validation set from the Pile-train split, with non-mentioned hyperparameters kept as the default. Especially, we filter out all the input sentences to the Llama 3.2-1B with a token length less than 1024 to keep the length of hidden state sets among all positions aligned. After the training, we run BAE again on all $262144$ samples before for binary encoding $h_1$, and calculate the marginal entropy on these $h_1$ as mentioned in Eq. 4.

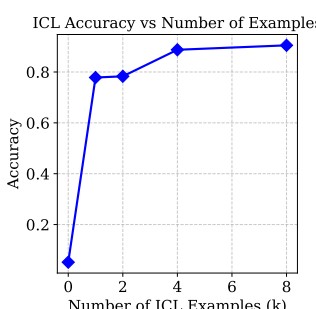

Figure 7: ICL accuracy calculated on the whole generated datasets on various $k$.

### A.3  TRACING IN-CONTEXT LEARNING INFERENCE BY ENTROPY (§4.3)

**Introduction of ICL and Input Format.** ICL (Radford et al., 2019; Dong et al., 2022) typically utilizes concatenations of input($x$)-answer($y$) pairs (with amount $k$, called demonstrations) to define a task, and requires the LM to generate the answer of the last input ($x_q$, called query) similar to the demonstrations. The inputs of ICL are built like $[x_1, y_1, x_2, y_2, \ldots, x_k, y_k, x_q]$. In practice, the input samples used in the experiments are built on the toolkit StaICC (Cho & Inoue, 2025), similar to the instance below:

---

sentence: a genuinely moving and wisely unsentimental drama . sentiment: positive
sentence: laughs – sometimes a chuckle , sometimes a guffaw and , to my great pleasure , the occasional belly laugh sentiment: positive
sentence: that the entire exercise has no real point sentiment: negative
sentence: 90 punitive minutes of eardrum-dicing gunplay , screeching-metal smashups , and flaccid odd-couple sniping sentiment: negative
sentence: amid the new populist comedies that underscore the importance of family tradition and familial community sentiment: positive
sentence: freakshow sentiment: negative
sentence: a taste for the quirky sentiment: positive
sentence: rustic , realistic , and altogether creepy sentiment: positive
sentence: assured direction and complete lack of modern day irony sentiment:

---

Figure 8: An example of ICL input in experiments of §4.3 with 8 demonstrations.

**Hyperparameters.** We sample $262144$ ICL input samples with $k \in \{0, 1, 2, 4, 8\}$, and extract the hidden states of the last token (i.e., ":") from layer $\{0, 2, 4, 6, 8, 10, 12, 14, 16\}$, with $209715$ samples for the training, and the remaining for validation. After the training, we run BAE again on all $262144$ samples before for binary encoding $h_1$, and calculate the marginal entropy on these $h_1$ as mentioned in Eq. 4.

**Accuracy.** As supplementary information, we test the classification accuracy of ICL on the given settings as shown in Fig. 7.

## A.4 ComSem Evaluation on BAE and SAE Variants (§5.3)

**Baselines.** As shown in Table 1, we utilize 4 baselines with implementation details shown below:

- **Sparse Autoencoder** (ReLU SAE). The feedforward calculation of ReLU SAE is defined as:

$$\hat{h_0} = \text{ReLU}(h_0 W_{\text{in}}) W_{\text{out}} + b. \tag{7}$$

  During the training, the $L_1$ normalization is applied to the inner activation $\text{ReLU}(h_0 W_{\text{in}})$:

$$\mathcal{L}(H_0) = \mathcal{L}_r(H_0) + \alpha \sum_{h_0 \in H_0} \|\text{ReLU}(h_0 W_{\text{in}})\|_1, \tag{8}$$

  the $\alpha$ is the normalization factor, defaulted to $10^{-7}$.

- **Top-$k$ SAE** (Bussmann et al., 2024). The feedforward calculation of Top-$k$ SAE is defined as:

$$\hat{h_0} = \text{Top}_k(h_0 W_{\text{in}}) W_{\text{out}} + b. \tag{9}$$

  The $\text{Top}_k$ function retains only the $k$ largest elements in place, setting all others to zero. During the training, only the regression loss is utilized:

$$\mathcal{L}(H_0) = \mathcal{L}_r(H_0), \tag{10}$$

  the $k$ is defaulted to 15.

- **Gated ReLU SAE** (Rajamanoharan et al., 2024a). The feedforward calculation of Gated ReLU SAE is defined as:

$$\hat{h_0} = \text{GateReLU}_\gamma(h_0 W_{\text{in}}) W_{\text{out}} + b. \tag{11}$$

  The $\text{GateReLU}_\gamma$ function is a thresholded variant of the ReLU, defined as:

$$\text{GateReLU}_\gamma(x) = \begin{cases} x, & x > \gamma, \\ 0, & \text{otherwise}, \end{cases} \tag{12}$$

  where $\gamma$ is a tunable gating parameter, defaulted to be 0.5. During the training, the $L_1$ normalization is applied to the inner activation $\text{GateReLU}_\gamma(h_0 W_{\text{in}})$:

$$\mathcal{L}(H_0) = \mathcal{L}_r(H_0) + \alpha \sum_{h_0 \in H_0} \|\text{GateReLU}_\gamma(h_0 W_{\text{in}})\|_1, \tag{13}$$

- **Transcoder** (Dunefsky et al., 2024). The feedforward calculation of Transcoder is defined as:

$$h_2 = \text{ReLU}(h_0 W_{\text{in}}) W_{\text{out}} + b. \tag{14}$$

  Where the $h_0$ is the input towards the $(l-1)$-th MLP block[9]. During the training, the $h_2$ is aligned to the output of the $(l-1)$-th MLP block, i.e., the hidden state of layer $l$, and $L_1$ normalization is applied to the inner activation:

$$\mathcal{L}(H_0) = \frac{1}{n_b} \|H_2 - H_l\|_2 + \alpha \sum_{h_0 \in H_0} \|\text{GateReLU}_\gamma(h_0 W_{\text{in}})\|_1, \tag{15}$$

  due to the significant difference in methodology with SAE variants, we do not regard Transcoder as a major comparison object.

**Parameters of BAE/SAE Training.** We sample $n = 8243323$ hidden state vectors from the specific layers of Llama 3.2-1B on the Pile-train split, with 6594658 as the training samples for BAE/SAE, and the remaining for the validation. The autoencoders are trained for 200 epochs, with $\alpha_e = 0$ in the first 50 epochs.

We visualize the burstiness-based activation magnitude calculation of BAE in Fig. 10.

---

[9]We note that the $(l-1)$-th MLP block produces the hidden states of layer $l$.

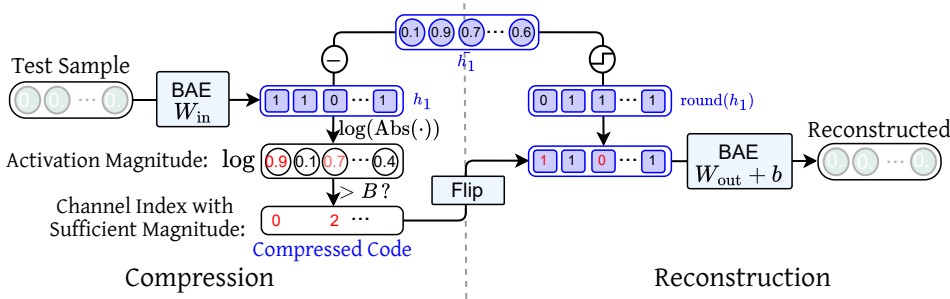

Figure 9: Diagram of hidden state compression and reconstruction utilizing BAE.

**Parameters of ComSem.** The pseudocode of ComSem is shown in Algorithm 1, with default parameters $N = 8192$, $n_I = 5$, $n_T = 8$, $k = 10$. The evaluations are conducted on the train split of BookCorpus, which is different from the training set of BAE/SAEs (Pile). The $\bar{h}_1$ is averaged among all the input instances of the

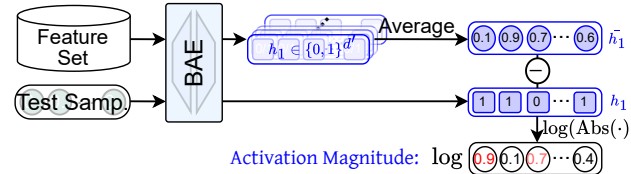

Figure 10: Burstiness-based activation magnitude calculation process described in §5.2.

evaluation set. The instructions for calling the backend LLMs are shown in Fig. 20 and 21, with queries' format the same as the given examples. We utilize GPT-4.1 and GPT-4.1-mini (Achiam et al., 2023) as the backend LM.

### A.5 VECTOR COMPRESSION BY BAE

As shown in Fig. 9, the compression of a hidden state vector $h_0$ proceeds as follows:

1. Given the input $h_0$ to be compressed, we first compute its binary encoding $h_1$ by applying the input projection $W_{\text{in}}$ and binarize function $\Gamma$ of a BAE, and then measure the burstiness $\beta$ with respect to the prior distribution $\bar{h}_1$, as described in §5.2 and Fig. 10.

2. Given a threshold[10] $B \in [\log 0.5, 0]$, we bookkeep only the channel indices $i$ for which $\beta_i > B$, using these indices as the compressed representation of $h_0$.

3. To reconstruct $h_0$ from the bookkept indices, we round $\bar{h}_1$ to a binary vector, flip the bits of $\text{round}(\bar{h}_1)$ at the stored indices, and then pass the flipped binary vector through $W_{\text{out}}$ (also the bias term) to obtain the recovered $h_0$.

**Hyperparameters & Experiment Settings.** We sample 117864 hidden state vectors from the specific layers on BookCorpus, then utilize the BAE trained on Pile for the compression, following the processing above. The $\bar{h}_1$ is averaged among all the input instances of the compressed set.

## B INTERPRETABILITY SCORE DISTRIBUTION: SELECTING A BETTER INTERPRETABLE FEATURE SET

We visualize the ComSem interpretability score distribution among features of BAE and baselines in Fig. 22 - 26, where compared to all the baselines, BAE significantly increased the amount of both high-score and low-score features. As shown in Appendix C, these low-score features are *non-literal*, that is, the semantics of these features can not be interpreted by the token inputs or outputs in ComSem, so that, despite serving as an effective proof of BAE's stronger extraction capability, harm the average interpretability score among all the activated features.

---

[10]Notice that the threshold greater than 0 is trivial, given the $h_1 \in \{0,1\}^{d'}$, $\bar{h}_1 \in [0,1]$, so that all the elements of $\log |h_1 - \bar{h}_1| \leqslant 0$. And burstiness less than $\log 0.5$ causes wrong flip of bits in Step 3.

Feature index: #111
Feature interpretation: "indefinite pronouns and quantifiers"
Interpretability score: 0.75

Activate tokens & contexts:
#1: "but just **one** look at a minion sent him practically catatonic ."
#2: "while **some** might look on her as having a mark against her character being an unwed mother , she had lived a relatively non-rebellious life ."
#3: "one **thing** her mother prided herself on was being on time and lending a hand ."
#4: "while it was good that he had so **many** male role models , she only hoped he had n't inherited too much of his father 's personality ."
#5: "after megan watched them disappear in the crowd of family and friends waiting in the church alcove , she bypassed **everyone** by turning right and heading down the hallway ."
#6: "more than **anything** , she wanted to have a little fun herself ."
#7: "tell me **what** ?"
...

Figure 11: Case analysis for feature 111 "indefinite pronouns and quantifiers" with interpretability score 0.75 in Llama 3.2-1B Layer 14.

However, as shown in the curves of Fig. 22 - 26 visualizing the average interpretability scores with all the previous sorted features, the curves of BAE are located above the baseline models in most cases. This indicates that when we remove some low-score features, BAE will achieve the highest average interpretability scores and also the number of interpretable features. This is a benefit of BAE's extensive feature extraction capability, which means that compared to baseline models that extract a limited number of features, BAE has space to perform trade-offs on both the number of features and interpretability scores, while ensuring that both are optimal, as shown in Table 3 with some of the low-score BAE features ignored.

## C   CASE ANALYSIS: EXTRACTED FEATURES

In this section, we observe several features along with their corresponding tokens and contexts where the investigated feature is activated, from the BAE trained on Layer 14 hidden states of Llama 3.2-1B. We observe in the category of these high-score and low-score features, and find that the low-score features are more folded and implicit in the hidden states, so it is harder to extract.

### C.1   HIGH-SCORE FEATURES

We first list some features with high interpretability scores, along with their corresponding activation tokens and contexts, as shown in Fig. 11 - 15. Among these high-score cases, we observe that: for the ComSem, it is easy to correctly identify semantic similarities grounded in the natural semantics of single tokens (e.g., different variants of the same word (Fig. 15), nouns describing objects of the same cat-

Table 3: Evaluation of BAE with low-score features ignored and baselines.

| Feat. Source | Model | Feature Activated | ComSem$_{4.1}$ | |
|---|---|---|---|---|
| | | | FI# | Score |
| Llama 3.2-1B Layer 11 $d = 2048$ | ReLU SAE | 2065 | 1380 | 0.260 |
| | Top-k SAE | 3417 | 2684 | 0.452 |
| | Gated ReLU SAE | 1226 | 1026 | 0.557 |
| | ReLU SAE Resc. | 744 | 482 | 0.278 |
| | TransCoder | 1794 | 1090 | 0.239 |
| | **BAE** (top 3500) | **3500** | **3500** | **0.574** |
| Llama 3.2-1B Layer 14 $d = 2048$ | ReLU SAE | 2528 | 1600 | 0.217 |
| | Top-k SAE | 2702 | 2004 | 0.418 |
| | Gated ReLU SAE | 2948 | 2250 | 0.435 |
| | ReLU SAE Resc. | 3962 | 2661 | 0.274 |
| | TransCoder | 3401 | 2166 | 0.267 |
| | **BAE** (top 3970) | **3970** | **3970** | **0.496** |
| Llama 3.2-3B Layer 20 $d = 3072$ | ReLU SAE | 1923 | 1289 | 0.312 |
| | Top-k SAE | 3234 | 2508 | 0.425 |
| | Gated ReLU SAE | 4628 | 3580 | 0.437 |
| | ReLU SAE Resc. | 2122 | 1451 | 0.294 |
| | TransCoder | 5508 | 3424 | 0.257 |
| | **BAE** (top 5700) | **5700** | **6805** | **0.544** |

egory (Fig. 12, 13, 14), or function words serving similar grammatical roles (Fig. 11)). We infer that: (1) The similarities in the hidden states of such tokens can be easily extracted from the residual stream based on the embedding vectors. Also, (2) the similarities in these activated tokens are clear towards the backend LLMs. So that these features can be interpreted into natural language interpretation from the plain tokens by LLMs appropriately.

---

Feature index: #1045
Feature interpretation: "numerical quantifiers and ordinals"
Interpretability score: 0.75

Activate tokens & contexts:
#1: "**one** day when they had their own place again , she would get him a dog ."
#2: "sean acknowledged her with a **two** finger salute before cranking up and pulling down the driveway ."
#3: "**one** thing her mother prided herself on was being on time and lending a hand ."
#4: "**one** time ."
#5: "but that was **six** months ago ."
#6: "aidan introduced him to his **four** sisters and their husbands ."
#7: "after exchanging hugs with emma and reassuring her at least **twenty** times that she would be fine and that she did need to go home , pesh led megan out the front door ."
...

---

Figure 12: Case analysis for feature 1045 "numerical quantifiers and ordinals" with interpretability score 0.75 in Llama 3.2-1B Layer 14.

---

Feature index: #2289
Feature interpretation: "verbs of visual attention or perception"
Interpretability score: 0.875

Activate tokens & contexts:
#1: "but just one **look** at a minion sent him practically catatonic ."
#2: "each time she **looked** into mason 's face , she was grateful that he looked nothing like his father ."
#3: "megan asked , g**azing** from noah 's...[10]"
#4: "**peek**ing out from the covering , she saw emma was wearing her signature color , green ."
#5: "of course , he 'd probably argue that while the gown might not have held up , he still **looks** fabulous and much younger than his age ."
#6: "megan **glanced** between the two of them ."
#7: "she **stared** into his face before she responded ."
...

---

Figure 13: Case analysis for feature 2289 "verbs of visual attention or perception" with interpretability score 0.875 in Llama 3.2-1B Layer 14.

## C.2 LOW-SCORE FEATURES

Also, we list some features with low interpretability scores, as shown in Fig. 16 - 19. Among these low-score features, we summarize the characteristics of these features: (1) **Subwords from the tokenizer.** As shown in Fig. 17, 19, these subword tokens share no obvious commonality even for humans, and they greatly confuse the LLM, leading it to produce irrelevant interpretations. This illustrates the drawback of token-based interpretation, as noted in our limitations. (2) **Function words with similar context.** As shown in Fig. 16, 18, these function-word tokens, despite appearing in similar contextual environments (often originating from the same sentence or even the same region within a sentence), lack surface-level commonality and are therefore difficult to interpret by ComSem. This highlights the urgency of interpreting directly from the decoded feature vectors instead of original tokens. Moreover, from this phenomenon, a further hypothesis is that function words without clear semantics may be more likely to concentrate contextual information during contextualization. (3) **Character level similarity.** As shown in Fig. 19, the tokens activated on feature 5949 all end with "em", according to previous works (Zhang & He, 2024; Fu et al., 2024), backend LLM may struggle to process such inputs, causing a low interpretability score.

---

[10]To comply with the requirements of the ethics review, we have redacted potentially offensive or NSFW content from these input cases.

Feature index: #7700
Feature interpretation: "body part nouns"
Interpretability score: 1.0

Activate tokens & contexts:
#1: "when he started to whine , she shook her **head** ."
#2: "he grinned and then happily dodged her mother 's **arms** for her father 's instead , which made megan smile ."
#3: "noah momentarily stopped sucking on the bottle to flash a quick smile , which warmed megan 's **heart** ."
#4: "emma rolled her **eyes** ."
#5: "casey gasped as her hand flew to her **chest** dramatically ."
#6: "casey tapped her **chin** with her index finger "
#7: "megan pursed her **lips** at the prospect ."
...

Figure 14: Case analysis for feature 7700 "body part nouns" with interpretability score 1.0 in Llama 3.2-1B Layer 14.

Feature index: #8183
Feature interpretation: "modal auxiliary verbs expressing ability or possibility"
Interpretability score: 1.0

Activate tokens & contexts:
#1: "as she went to the couch and picked him up , she **could** n't help finding it amusing that out of everyone he was going to see today , he was most excited about being with aidan and emma 's black lab , beau ."
#2: "sadly , she **could** n't say that her first love was davis , mason 's father ."
#3: "considering i have two younger brothers , i think i **can** handle him ."
...

Figure 15: Case analysis for feature 8183 "modal auxiliary verbs expressing ability or possibility" with interpretability score 1.0 in Llama 3.2-1B Layer 14.

### C.3 SCALE OF HIGH-SCORE FEATURES AND LOW-SCORE FEATURES

In this section, we argue that: compared to high-scoring features, low-scoring features are embedded in more fine-grained structures of the hidden states, making them difficult to cluster with UMAP. Consequently, extracting such features requires stronger extraction capability. Compared to the baselines, BAE captures more of these features, confirming its superior feature extraction ability.

In detail, we conduct UMAP (McInnes et al., 2018), a dimensionality reduction based on the macroscopic adjacency structure from Euclidean distance, on hidden states of all the tokens from the sentences whose token activates the investigated feature, as shown in Fig. 40 for 3 high-score features, and Fig. 41 for 3 low-score features, where an important observation is that: the hidden states activated by high-scoring features form clear clusters, suggesting macroscopic spatial similarity. In contrast, the hidden states activated by low-scoring features are more dispersed overall, exhibiting similarity only within certain subspaces (i.e., along the corresponding rows of the activated features in $W_{out}$). That is, *the low-score features are more folded and implicit*, so that harder to extract.

## D AUGMENTATION EXPERIMENTS AND RESULTS

### D.1 ACTIVATION MAGNITUDE VISUALIZATION OF BAE AND SAE

We visualize the activation magnitude of each inputted $h_0$ (vertical axis) on each channel (horizontal axis) to observe the sparsity of BAE (left sub-figures) and SAE (right sub-figures) as shown in Fig. 33 - 36. Besides the clear sparsity of BAE, the visualization of SAE shows vertical stripes,

Feature index: #514
Feature interpretation: "common auxiliary verbs and determiners"
Interpretability score: 0.0

Activate tokens & contexts:
#1: "he 'd seen the movie almost by mistake , considering he was a little young for the pg cartoon , but with older cousins , along with her brothers , mason was **often** exposed to things that were older ."
#2: "he 'd seen the movie almost by mistake , considering he was a little young for the pg cartoon , but with older cousins , along with her brothers , mason was often exposed to things that **were** older ."
#3: "she liked to think being surrounded by adults and older kids was one reason why he **was** a such a good talker for his age ."
#4: "she liked to think being surrounded by adults and older kids was one reason why he was a such **a** good talker for his age ."
#5: "it was only his build that he **was** taking after his father ."
#6: "while it had been no question that she wanted him as godfather for mason , she had been extremely honored when he and his wife , emma , had asked her to be their son , noah '**s** , godmother ."
#7: "while it had been no question that she wanted him as godfather for mason , she had been extremely honored when he and his wife , emma , had asked her to be their son , noah 's **,** godmother ."
#8: "i plan on spoiling noah rotten and corrupting him as **only** a good auntie can do !"
#9: "i plan on spoiling noah rotten and corrupting him as only **a** good auntie can do !"
#10: "i plan on spoiling noah rotten and corrupting him as only a **good** auntie can do !"
...

Figure 16: Case analysis for feature 514 "common auxiliary verbs and determiners" with interpretability score 0.0 in Llama 3.2-1B Layer 14.

Feature index: #2410
Feature interpretation: "abbreviations or truncated forms of words"
Interpretability score: 0.0

Activate tokens & contexts:
#1: "he reminds me of that bollywood actor john **ab**raham , " casey said ."
#2: "as he eyed the massive statue of jesus , he fidgeted **abs**ently with his tie ."
#3: "he was taken **ab**ack by her words and the passion with which she delivered them ."
#4: "her eyes **fr**antically scanned the room ."
#5: "her gaze flicked down to the tan , **mus**cled arm ."
#6: "he welcomed the pain as she went over the edge , **conv**ulsing and screaming against his hand ."
#7: "instead , she just enjoyed being close to him , the feel of his hand on her , his strong , **mus**cled thighs beneath her ."
...

Figure 17: Case analysis for feature 2410 "common auxiliary verbs and determiners" with interpretability score 0.0 in Llama 3.2-1B Layer 14.

indicating consistently high-activated channels, which are absent in BAE. This suggests that utilizing global normalization for sparsity, and calculating burstiness as the activation magnitude effectively suppresses the dense activations seen in SAE.

## D.2 MORE CASES FOR FIG. 3

We examine the training dynamics of BAE with and without the entropy objective ($\mathcal{L}_e$) under additional settings, extending the observations from Fig. 3. The results, shown in Fig. 27 - 32, reveal

Feature index: #3479
Feature interpretation: "adverbs and pronouns used as modifiers or objects"
Interpretability score: 0.0

Activate tokens & contexts:
#1: "each time she looked into mason 's face , she was grateful that he looked **nothing** like his father ."
#2: "with a sigh , she stepped into the dress and slid **it** over her hips .'
#3: "while it boasted a sweetheart neckline , the hemline **fell** just below her knees ."
#4: "while it boasted a sweetheart neckline , the hemline fell **just** below her knees ."
#5: "we 'll see you there in **just** a few , " her mother called ."
#6: "with her grades **already** in the toilet , she was unprepared for the emotional breakdown she experienced when carsyn broke up with her ."
#7: "with her grades already in **the** toilet , she was unprepared for the emotional breakdown she experienced when carsyn broke up with her ."
...

Figure 18: Case analysis for feature 3479 "adverbs and pronouns used as modifiers or objects" with interpretability score 0.0 in Llama 3.2-1B Layer 14.

Feature index: #5949
Feature interpretation: "tokens related to trembling or shaking actions"
Interpretability score: 0.0

Activate tokens & contexts:
#1: "her chin **trem**bled as she replied , " i want that for my son . ""
#2: "her lip **trem**bled .'
#3: "her chin **trem**bled before big , fat tears slid down her cheeks ."
#4: "her body **trem**bled slightly at his words ."
#5: "**lem** me guess , you bribed him sexually to get him to give up a perfectly good saturday watching two kids who are under two ."
...

Figure 19: Case analysis for feature 5949 "tokens related to trembling or shaking actions" with interpretability score 0.0 in Llama 3.2-1B Layer 14.

that a larger rank leads to more complex training behavior, while incorporating the entropy objective reshapes the dynamics into a smoother and more monotonic form.

D.3 MORE CASES FOR FIG. 6

We extend the investigation scope of Fig. 6, where the activation frequency of each feature is visualized, as shown in Fig. 37 - 39. The conclusion derived from these augmentation results is consistent with the main body.

Instruction:
I will provide a set of tokens along with their positions (this position may vary depending on the tokenizer) and the surrounding context. Please describe what these tokens have in common using concise expressions such as "date expressions", "words ending in 'ing'", or "adjectives".
Please choose the most specific term while ensuring commonality, and avoid using overly general terms like "words", "English tokens", "high-frequency English lexemes", or "phrases".
Non-semantic or non-linguistic terms such as "BPE Subword Token" are strictly prohibited. Any additional information, explaination, or context are strongly prohibited. Only return one phrase.

Example 1:
Token: "running" at position 3 in sentence: "She is running in the park."
Token: "eating" at position 3 in sentence: "He is eating an apple."
Token: "sleeping" at position 4 in sentence: "The baby is sleeping on the sofa."
Token: "jumping" at position 2 in sentence: "They are jumping over the fence."
Token: "talking" at position 3 in sentence: "We are talking about the project."
The commonality is: -ing verbs of human behavior

Example 2:
Token: "yesterday" at position 4 in sentence: "I went there yesterday."
Token: "last week" at position 5 in sentence: "She arrived last week."
Token: "in 1998" at position 6 in sentence: "They moved here in 1998."
Token: "last year" at position 5 in sentence: "We met last year."
The commonality is: past time expressions

Example 3:
Token: "happy" at position 4 in sentence: "She looks very happy today."
Token: "angry" at position 5 in sentence: "They were extremely angry about it."
Token: "sad" at position 4 in sentence: "He felt really sad after the call."
The commonality is: emotional adjectives

Example 4:
Token: "dog" at position 2 in sentence: "The dog barked loudly."
Token: "cat" at position 2 in sentence: "The cat chased the mouse."
Token: "bird" at position 2 in sentence: "The bird sang beautifully."
Token: "fish" at position 2 in sentence: "The fish swam gracefully in the tank."
The commonality is: animal nouns

(...)

Example 11:
Token: "sad" at position 4 in sentence: "He felt really sad after the call."
Token: "angry" at position 5 in sentence: "They were extremely angry about it."
Token: "negative" at position 4 in sentence: "She had a negative reaction to the news."
The commonality is: negative emotion adjectives

Now, please analyze the following tokens and their contexts:
(Test Sample)

Figure 20: Instruction text for interpreting the feature semantics.

Background:
I will provide a token, its position in the sentence, the surrounding context, and a candidate description of the token's role or type given the context.
Your task is to judge whether the given description accurately characterizes the token in its context.
Please respond with either:
- "Yes" (if the description is accurate), or
- "No" (if it is inaccurate)
Any additional information, explanation, or context are strongly prohibited. Only return "Yes" and "No".

Example 1:
Token: "running" at position 3 in sentence: "She is running in the park."
Candidate description: "present participle"
Answer: Yes

Example 2:
Token: "dog" at position 2 in sentence: "The dog barked loudly."
Candidate description: "adjective"
Answer: No

Example 3:
Token: "quickly" at position 4 in sentence: "He ran quickly toward the exit."
Candidate description: "manner adverb"
Answer: Yes

Example 4:
Token: "first" at position 4 in sentence: "This is the first time I have seen this."
Candidate description: "ordinal number"
Answer: Yes

Example 5:
Token: "to" at position 5 in sentence: "I want to go to the store."
Candidate description: "emotional verb"
Answer: No

Example 6:
Token: "looking" at position 3 in sentence: "She is looking forward to the event."
Candidate description: "verb related to oral communication"
Answer: No

(...)

Example 22:
Token: "fish" at position 2 in sentence: "The fish swam gracefully in the tank."
Candidate description: "noun describing an animal"
Answer: Yes

Now, please analyze the following tokens, their positions, contexts, and candidate descriptions:
(Test Sample)

Figure 21: Instruction text for testing the interpretation.

---

**Algorithm 1:** **Com**mon **Sem**antics-based Feature Interpretation and Evaluation (ComSem).

---

**Parameters:** $f : \mathbb{R}^d \to \mathbb{R}^{d'}$: Feature encoder, mapping original input $h_0$ to the **magnitude** of decomposed features $h'$ (e.g., encoding part of a typical SAE, or burstiness calculation for a BAE).

$\qquad\qquad$ $d'$: feature numbers.

$\qquad\qquad$ $n_I$: samples for generating interpretation.

$\qquad\qquad$ $n_T$: max test samples.

$\qquad\qquad$ $k$: top-? features that are seemed as activated.

$\qquad\qquad$ LM$(\cdot)$: backend LM client for generate the interpretation and test.

$\qquad\qquad$ $\text{Prompt}_{\text{Interp}}(\cdot)$: Prompt template for interpretating the features (described in Fig. 20).

$\qquad\qquad$ $\text{Prompt}_{\text{Test}}(\cdot)$: Prompt template for interpretating the features (described in Fig. 21).

**Input:** Set (of amount $N$) of token-hidden state of token-context triple: $\mathcal{S} = \{(t, h_0, c)^{(i)}, c = t_j\}_{i=1}^N$

**Initialization:** $\mathcal{D}_S = \{\}$: dictionary for each feature (length: $d'$), keeping the samples where the corresponding feature is activated.

$\qquad\qquad$ $\mathcal{D}_I = \{\}$: dictionary for each feature, keeping the interpretation and evaluation score.

$\qquad\qquad$ FA $= 0$: Number of feature activated.

$\qquad\qquad$ FI $= 0$: Number of interpretable features.

$\qquad\qquad$ Score $= 0$: Averaged interpretability score among the successfully interpreted features.

$\quad$ /* 1. Find the activated input samples for each feature channel. $\qquad\qquad\qquad\qquad\qquad$ */

1 **for** $i := 1$ *to* $n$ **do**

2 $\quad$ $a := f(h_0^{(i)})$ // Calculate activation magnitude of each feature channel.

3 $\quad$ $\mathcal{I} := \arg\max_k(a)$ // Find top-$k$ activated feature index.

4 $\quad$ **for** $j$ *in* $\mathcal{I}$ **do**

5 $\quad\quad$ $\mathcal{D}_S[j].\texttt{append}(t^{(i)},\ c^{(i)})$ // Bookkeep the activated sample for each feature channel.

6 $\quad$ **end**

7 **end**

$\quad$ /* 2. Get the interpretation for each feature channel. $\qquad\qquad\qquad\qquad\qquad\qquad\qquad$ */

8 **for** $i := 1$ *to* $d'$ **do**

9 $\quad$ **if** $\texttt{len}\,(\mathcal{D}_S[i]) \geqslant n_I + 1$ **then**

$\quad\quad$ // Query the LLM for the interpretation, only use $n_I$ samples, with the remaining ones for evaluation.

10 $\quad\quad$ $\mathcal{D}_I[i] := \{\texttt{"Interp"} = \text{LM}(\text{Prompt}_{\text{Interp}}(\mathcal{D}_S[i][1:n_I])),\ \texttt{"Activated"} = \texttt{True}\}$

11 $\quad\quad$ FA $:=$ FA $+ 1$

12 $\quad$ **else**

$\quad\quad$ // Reject the interpretation if samples are not sufficiently loaded.

13 $\quad\quad$ $\mathcal{D}_I[i] := \{\texttt{"Interp"} = \texttt{None},\ \texttt{"Activated"} = \texttt{False}\}$

14 $\quad$ **end**

15 **end**

$\quad$ /* 3. Evaluate the interpretation for activated feature channels. $\qquad\qquad\qquad\qquad$ */

16 **for** $i := 1$ *to* $d'$ **do**

17 $\quad$ **if** $\mathcal{D}_I[i][\texttt{"Activated"}]$ **then**

18 $\quad\quad$ $\mathcal{D}_I[i][\texttt{"Score"}] := 0$

19 $\quad\quad$ **for** $(t^{(j)}, c^{(j)})$ *in* $\mathcal{D}_S[i][n_I + 1 : n_I + n_T]$ **do**

$\quad\quad\quad$ // For each query in the test set (number bounded by the $n_T$), query the LLM for judging the matching of each input sample and its interpretation in True or False.

20 $\quad\quad\quad$ **if** $\text{LM}(\text{Prompt}_{\text{Test}}((t^{(j)}, c^{(j)}), \mathcal{D}_I[i][\texttt{"Interp"}]))$ **then**

21 $\quad\quad\quad\quad$ $\mathcal{D}_I[i][\texttt{"Score"}] = \mathcal{D}_I[i][\texttt{"Score"}] + 1$

22 $\quad\quad\quad$ **end**

23 $\quad\quad$ **end**

24 $\quad\quad$ $\mathcal{D}_I[i][\texttt{"Score"}] = \mathcal{D}_I[i][\texttt{"Score"}]/\texttt{len}\,(\mathcal{D}_S[i][n_I + 1 :])$

25 $\quad\quad$ Score $:=$ Score $+ \mathcal{D}_I[i][\texttt{"Score"}]$

26 $\quad\quad$ **if** $\mathcal{D}_I[i][\texttt{"Score"}] > 0$ **then**

27 $\quad\quad\quad$ FI $:=$ FI $+ 1$

28 $\quad\quad$ **end**

29 $\quad$ **end**

30 **end**

31 **return** $\mathcal{D}_I$, Score/FA, FA, FI

---

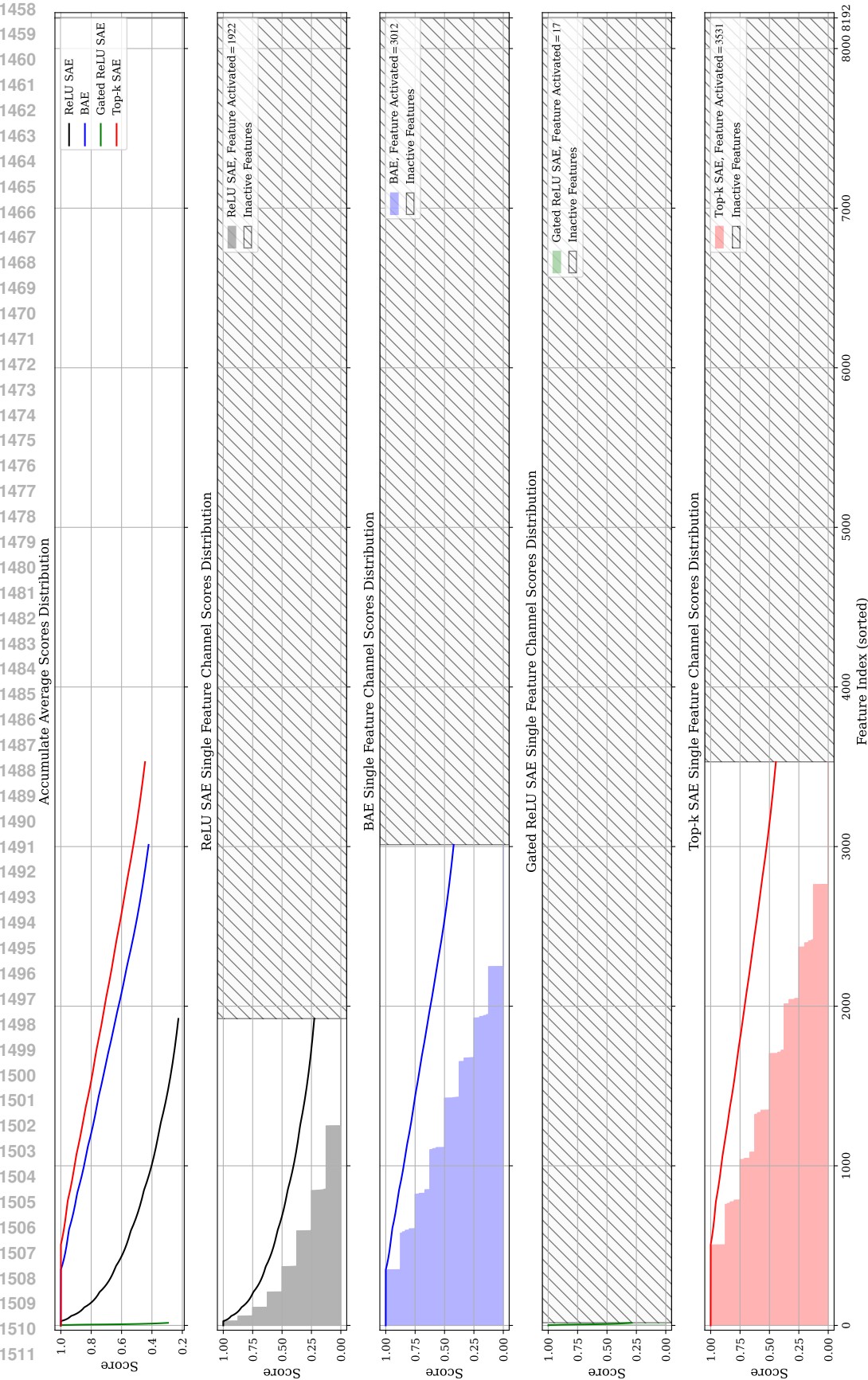

Figure 22: ComSem interpretability score distribution of each feature channel on layer 5 of Llama 3.2-1B. The colored areas represent the interpretability scores of individual channels (sorted), and the curves show the averaged interpretability scores across all channels previously.

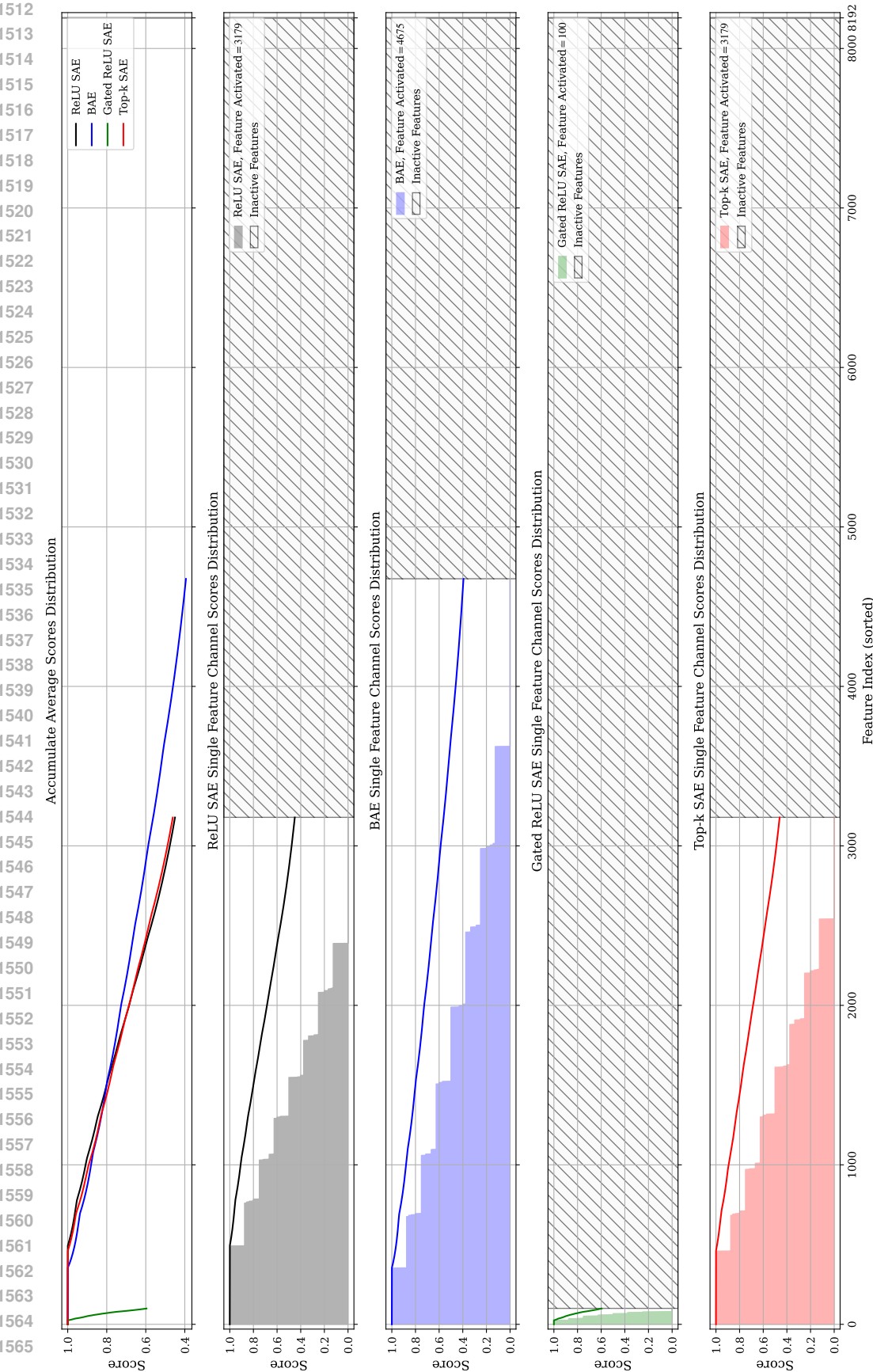

Figure 23: ComSem interpretability score distribution of each feature channel on layer 9 of Llama 3.2-1B.

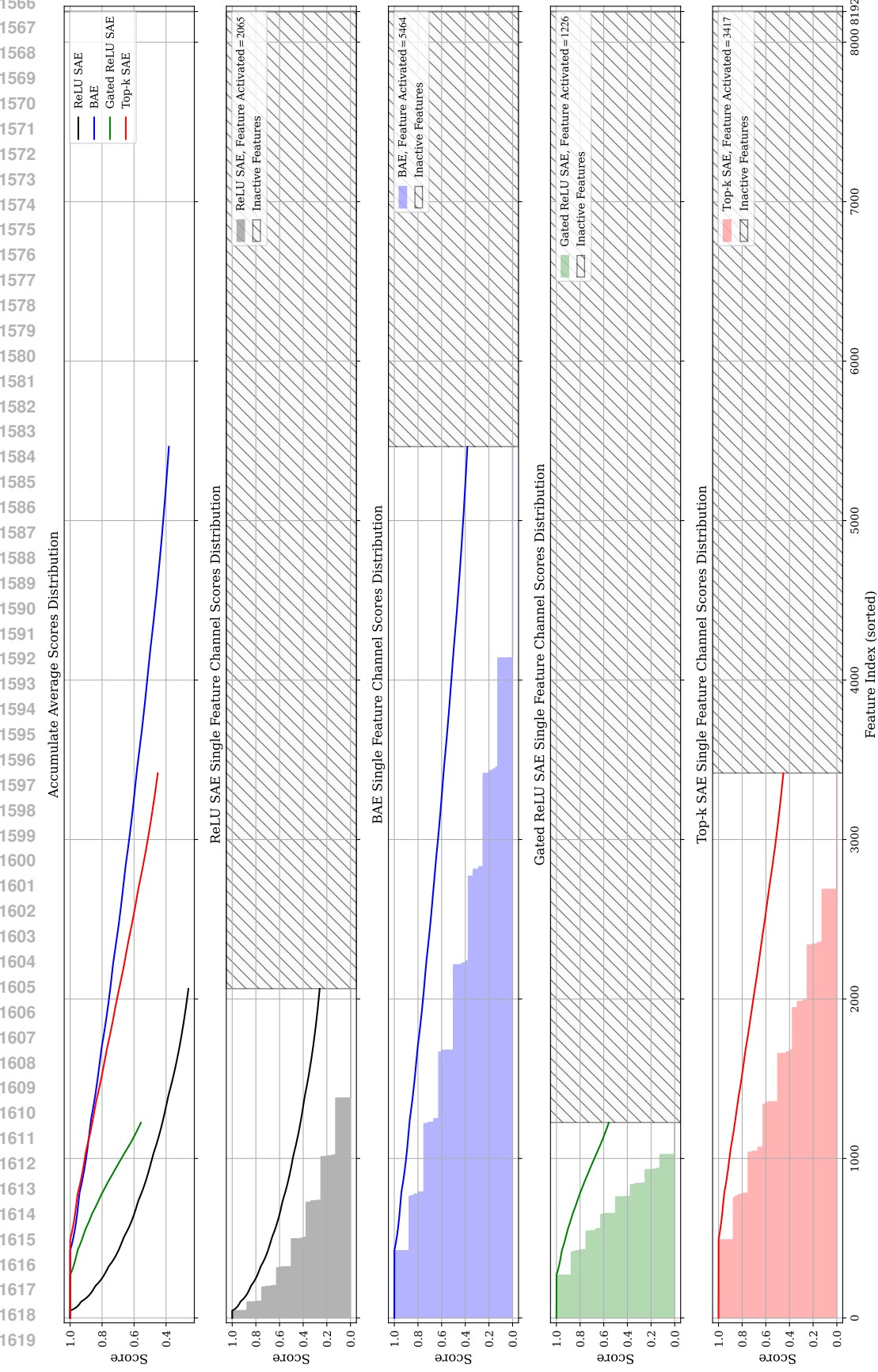

Figure 24: ComSem interpretability score distribution of each feature channel on layer 11 of Llama 3.2-1B.

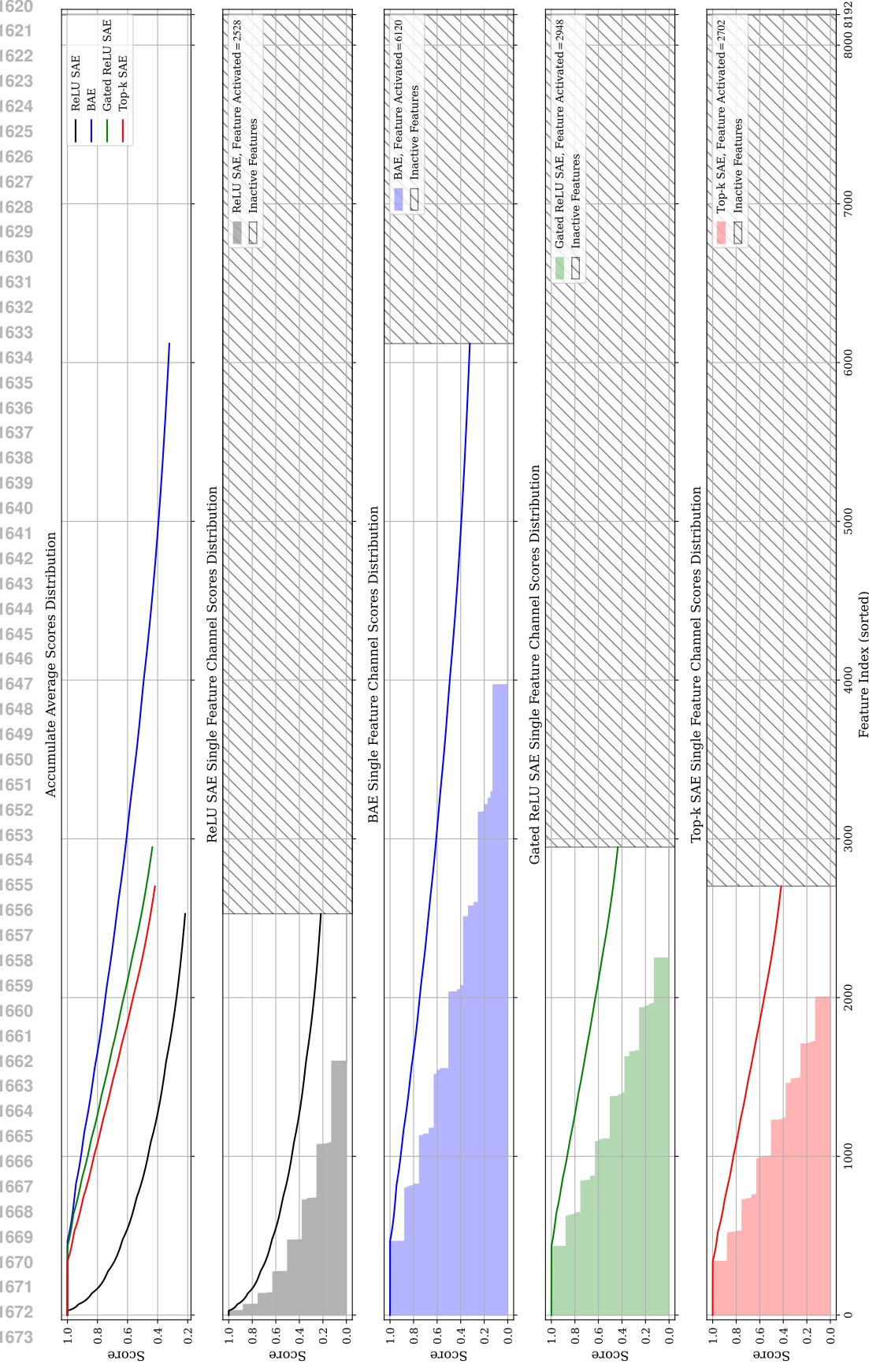

Figure 25: ComSem interpretability score distribution of each feature channel on layer 14 of Llama 3.2-1B.

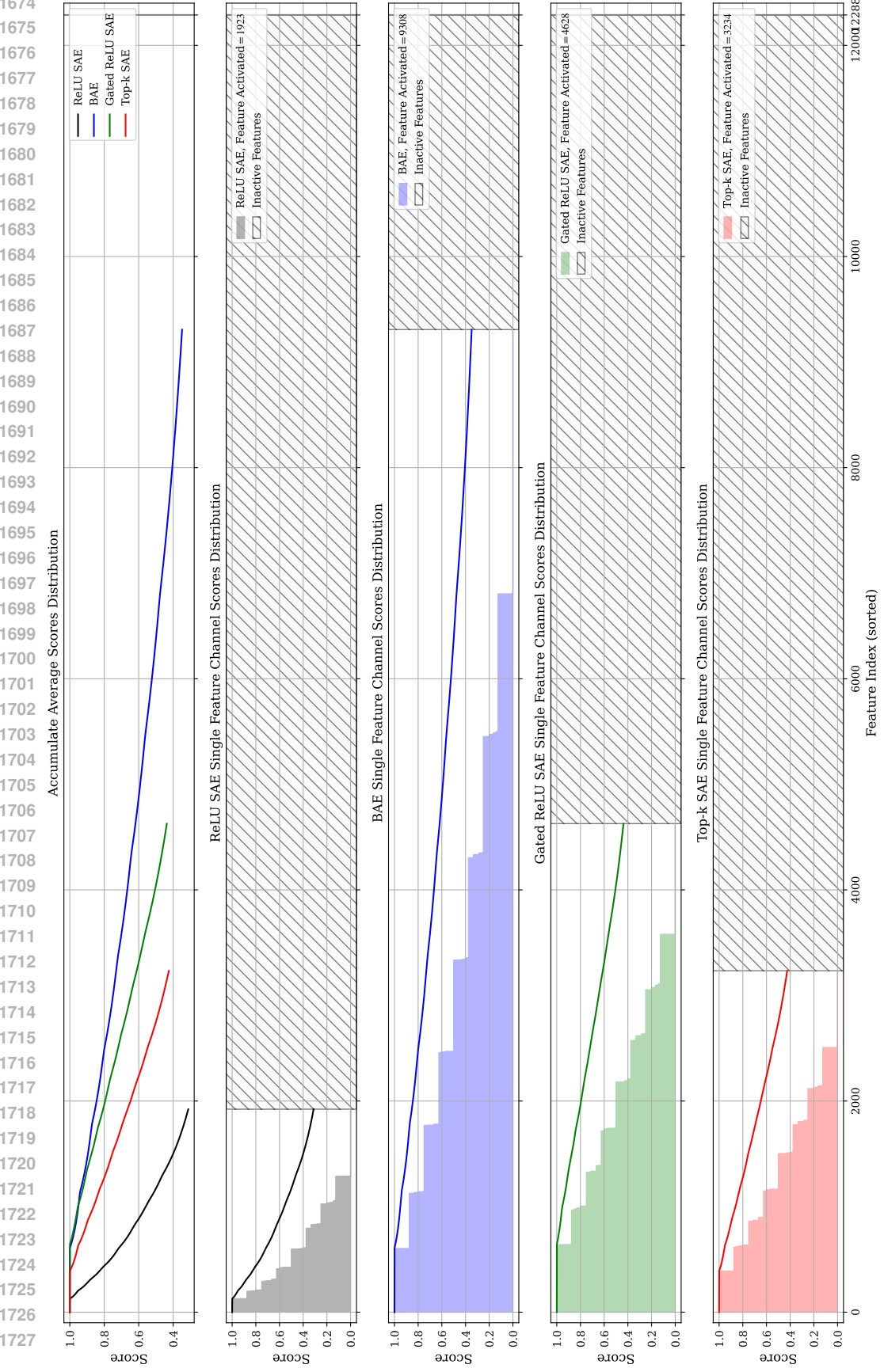

Figure 26: ComSem interpretability score distribution of each feature channel on layer 20 of Llama 3.2-3B.

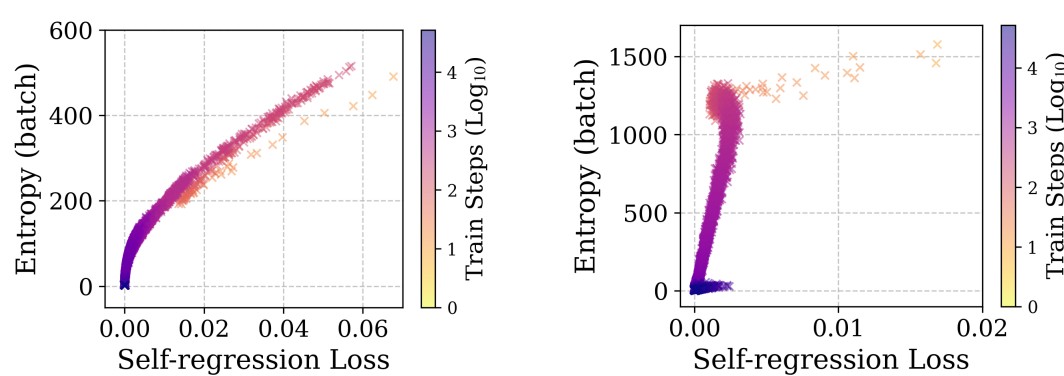

Figure 27: Augment result for Fig. 3 ($r = 2$, $d' = 2d$). (Left) with entropy objective, (right) without entropy objective.

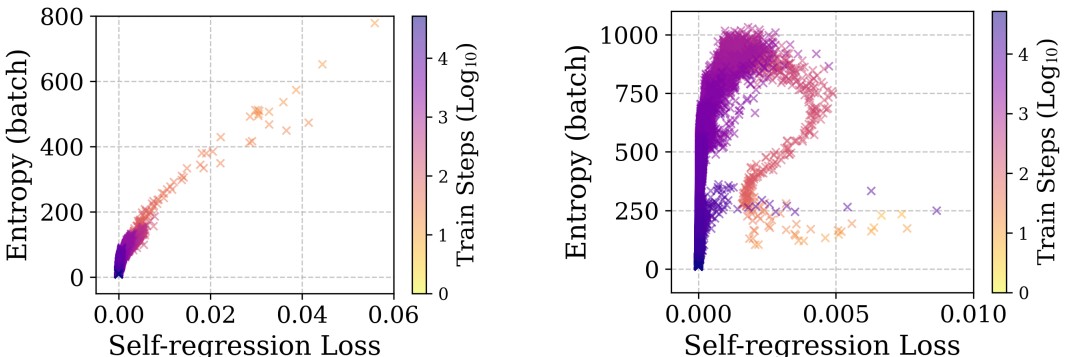

Figure 28: Augment result for Fig. 3 ($r = 8$, $d' = 2d$). (Left) with entropy objective, (right) without entropy objective.

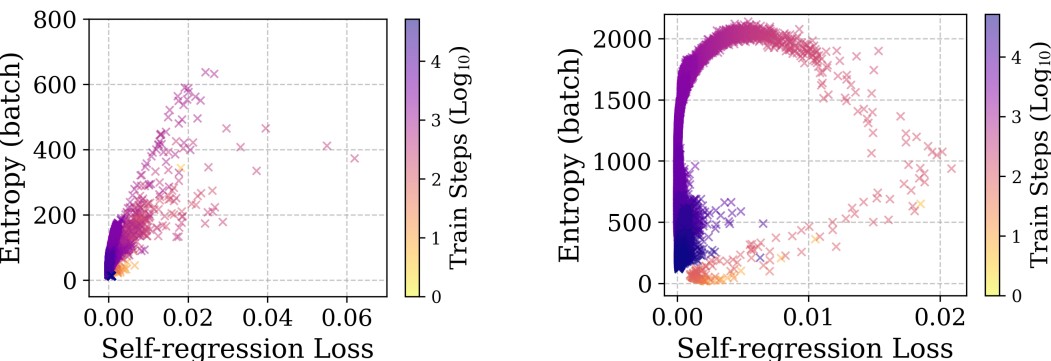

Figure 29: Augment result for Fig. 3 ($r = 16$, $d' = 4d$). (Left) with entropy objective, (right) without entropy objective.

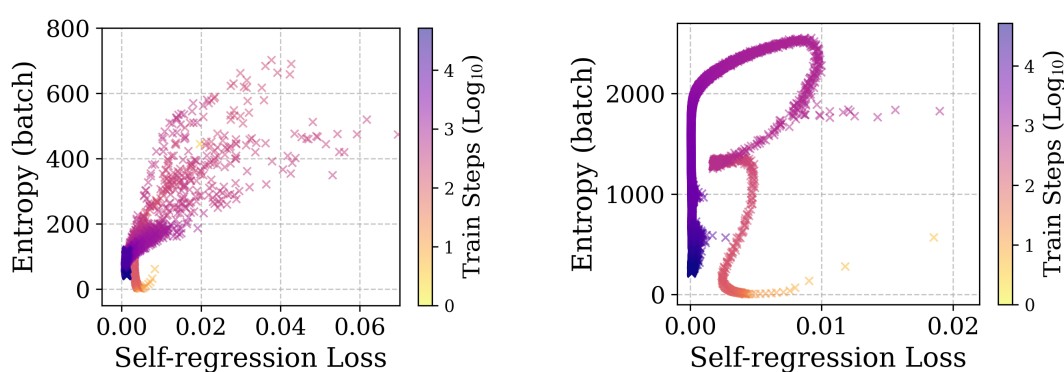

Figure 30: Augment result for Fig. 3 ($r = 32$, $d' = 4d$). (Left) with entropy objective, (right) without entropy objective.

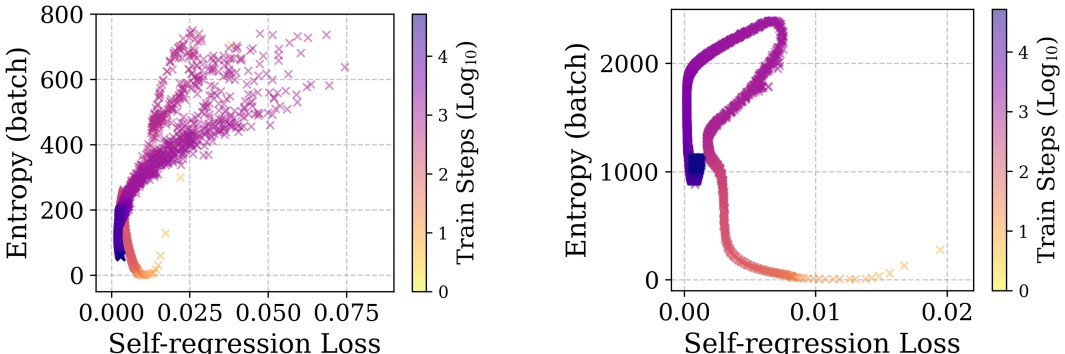

Figure 31: Augment result for Fig. 3 ($r = 64$, $d' = 4d$). (Left) with entropy objective, (right) without entropy objective.

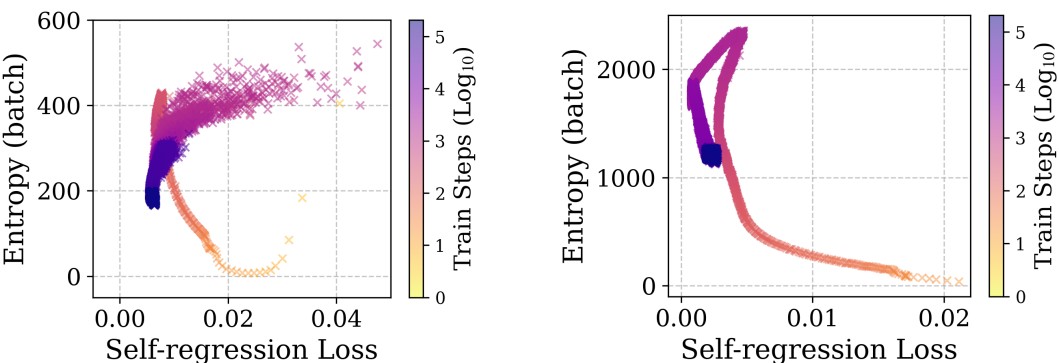

Figure 32: Augment result for Fig. 3 ($r = 128$, $d' = 4d$). (Left) with entropy objective, (right) without entropy objective.

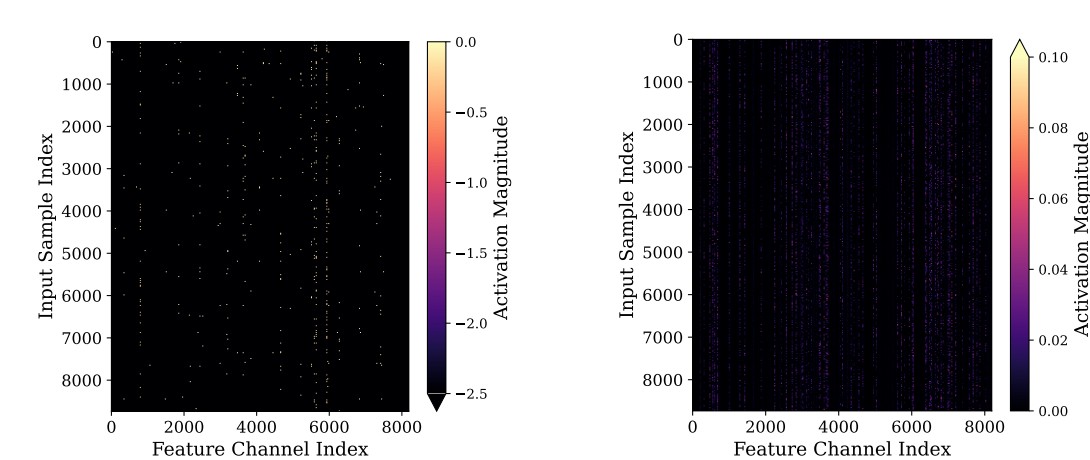

Figure 33: Activation magnitude visualization of (left) BAE and (right) SAE of layer 5.

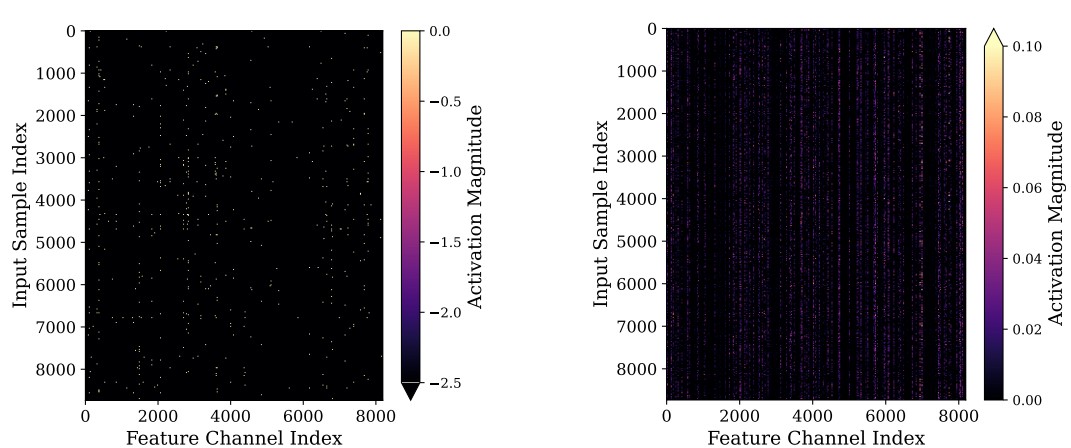

Figure 34: Activation magnitude visualization of (left) BAE and (right) SAE of layer 9.

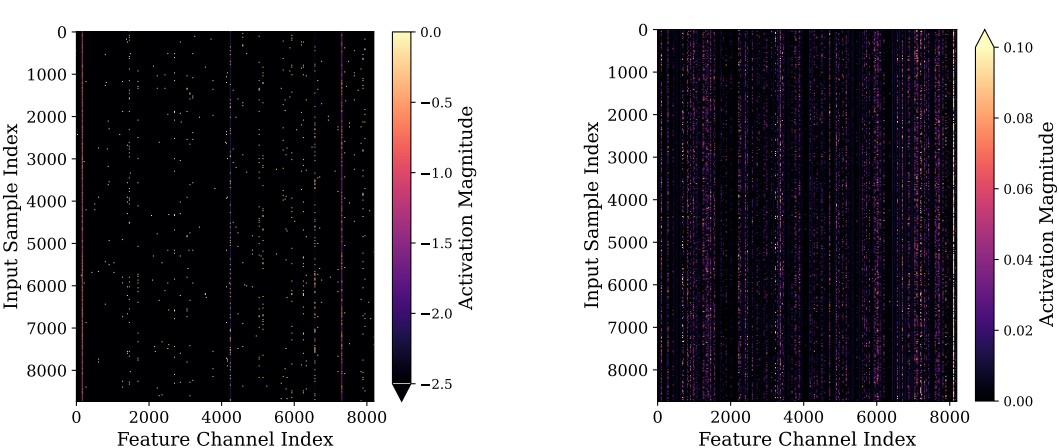

Figure 35: Activation magnitude visualization of (left) BAE and (right) SAE of layer 11.

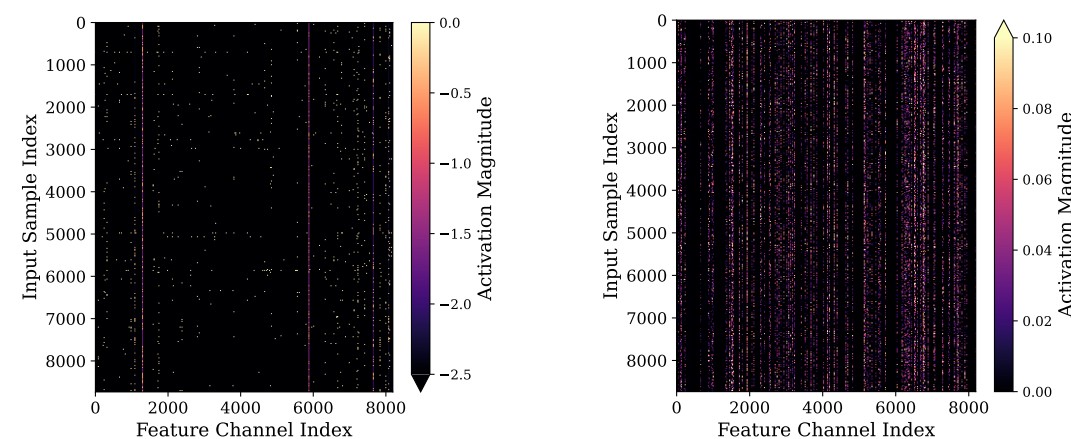

Figure 36: Activation magnitude visualization of (left) BAE and (right) SAE of layer 14.

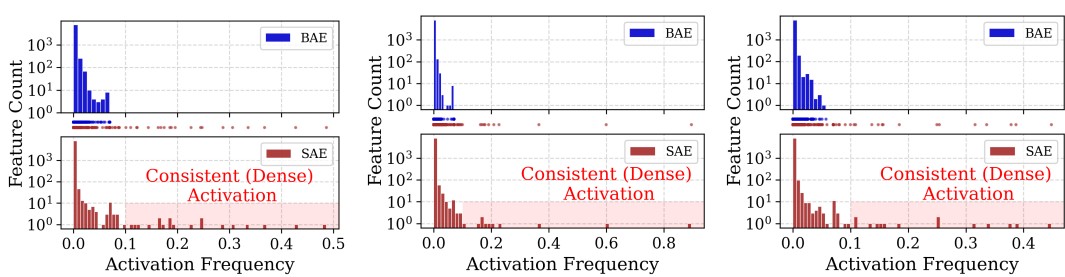

Figure 37: Augment for Fig. 6 on Layer 5 of Llama 3.2-1B.

Figure 38: Augment for Fig. 6 on Layer 9 of Llama 3.2-1B.

Figure 39: Augment for Fig. 6 on Layer 14 of Llama 3.2-1B.

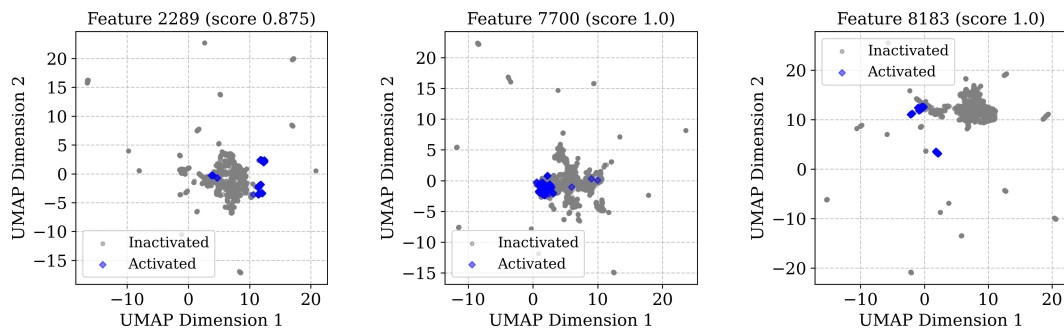

Figure 40: UMAP visualization for 3 high-score features with corresponding activated/inactivated hidden states on Layer 14 of Llama 3.2-1B.

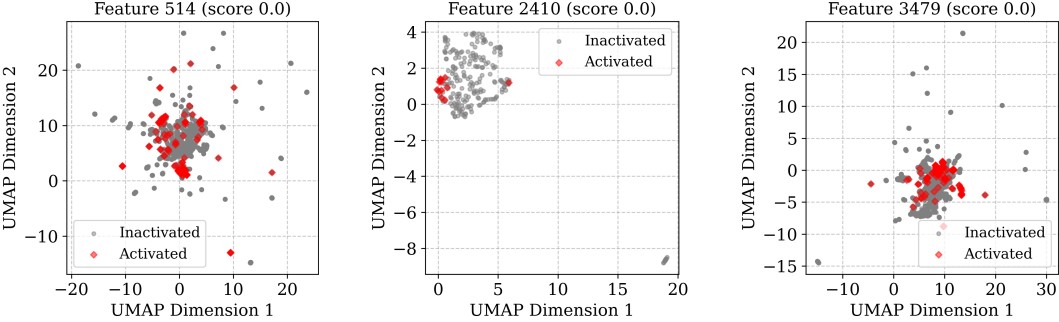

Figure 41: UMAP visualization for 3 low-score features with corresponding activated/inactivated hidden states on Layer 14 of Llama 3.2-1B.

