# OpenReview forum: "Binary Autoencoder for Mechanistic Interpretability of Large Language Models"
_ICLR.cc/2026/Conference — Submitted to ICLR 2026_

### Official Review · Reviewer_N6Vy · 2025-10-25

**Soundness:** 2
**Presentation:** 3
**Contribution:** 2
**Rating:** 4
**Confidence:** 4

**Summary:**

This paper presents a variant of Sparse Autoencoders (SAEs), named Binary Autoencoder (BAE), which enforces entropy-based loss on hidden activations and obtains binarized hidden activations. Authors conduct experiments to prove that BAEs can serve as a tool for analyzing information stored in LLM activations, and use BAEs to evaluate information in 1) normal sentence modeling, and 2) in-context learning. To interpret BAE features, the authors propose to use common semantics-based feature interpretation, and show BAEs have competitive interpretability with SAEs and their variants.

**Strengths:**

1. The paper is well-written and easy to follow. The figures are clear.
2. The idea of binary feature activations is novel. This is a promising method since it eliminates the activation of different magnitudes in SAEs.
3. BAEs provide a novel and performant method to estimate information of high-dimensional real-valued activations. Compared to probability density estimation, where the computational complexity is $O(C^d)$, BAEs provide a far more efficient method to estimate information. The conclusion they reached -- information increases with token location in normal sentence modeling, but decreases in in-context learning -- is interesting and inspiring.
4. BAEs show competitive interpretability with SAEs and their variants.

**Weaknesses:**

1. The definition of gradient estimation of $\Gamma$ in Eq. (6) seems to be wrong. For sigmoid function $S(x)$, the gradient should be $S'(x)=S(x) \\odot (1-S(x)$, but not $x\\odot(1-x)$. Also, this gradient estimation lacks some theoretical basis.
2. The synthetic directional benchmarking in Section 4.1 is in rather simple setting, where ground truth features are all mutually independent and have equal probabilities of activating and non-activating. This settings seem to be far from the real scenario of language modeling or other tasks. More complicated settings, such feature activating sparsity and correlation between features, should be introduced to better prove the entropy calculation ability of BAEs.
3. The ComSem method introduced in Section 5.1 seems to be incremental. Both ComSem and current method collect samples with significant activation magnitudes, and ask LLMs to provide explanation. The difference lies only in the evaluation part, where current method asks LLMs to simulate feature activation prediction, while ComSem gives the explanation and feature activations to LLMs, and let LLMs judge if the explanation is right. It lacks some evidence that ComSem's evaluation is more effective since LLMs may still misjudge it.
4. Some statistics on sparsity and reconstruction fidelity are provided in Figure 6 and Table 2. But common SAE metrics, such as L0 norm and explained variance, are not provided.
5. The tasks are restricted to two language models, Llama 3.2 1B and 3B. More models should be included to prove the generalizability of BAEs.
6. Typos:
   1. Line 055: "understang" should be "understanding".
   2. Line 206: $c\in 0,1^r$ should be $c\\in\\{0,1\\}^r$.

**Questions:**

1. What is the meaning of the $D[\Gamma(H_0W_\\text{in})]$ part of the entropy loss $\\mathcal{L}\_e(H_0)$ in Eq. (4)? I think the $H\_0W\_\text{in}$ is not a square matrix since batch size may not be equal to feature number. And why should the margin entropy be constrained?
2. Why should separate BAEs be trained on different token positions in Section 4.2 and 4.3? This is unnatural since SAEs are often trained on activations on all token positions.
3. The activation magnitude estimation in Section 5.2 is a bit unintuitive. In the BAE case, ${h_1}_j^{(i)}$ should be either 0 or 1, leading to $\beta_j^{(i)}$ also being binarized. Whether a feature at a token is considered significant is decided by whether more significant features exist to take up the top-$k$ quotas. Can this be clarified better?
4. SAE features of different feature activation magnitude may exhibit different meanings. Do this problem exists in BAEs?

---

### Official Review · Reviewer_Ah41 · 2025-10-26

**Soundness:** 1
**Presentation:** 2
**Contribution:** 3
**Rating:** 2
**Confidence:** 3

**Summary:**

This paper proposes a Binary Autoencoder (BAE) as an alternative to the Sparse Autoencoder (SAE) for mechanistic interpretability. Specifically, motivated by the observation that SAEs produce an excessive number of dense and 'dead' features , BAE is trained by constraining the minimal entropy of a minibatch  and imposing additional penalty constraints on sparse features. Concurrently, it binarizes the latent activations (dictionary indices)  to facilitate entropy computation. Subsequent, extensive experiments demonstrate the potential applications and capabilities of BAE.

**Strengths:**

The core idea of the paper—minimizing the entropy of binary activations over a minibatch —is highly novel. It fundamentally addresses the "global density" problem caused by SAE's reliance on sample-level $L_1$ sparsity , shifting instead to an information-theoretic, batch-level perspective to ensure global sparsity. Furthermore, the potential applications of BAE as an analytical tool, as demonstrated in several domains (e.g., tracing LLM inference and ICL), are intriguing. I appreciate the methodology of this paper and believe that BAE can indeed explore regions of mechanistic interpretability that remain uncharted.

**Weaknesses:**

1. I believe the binarization step, namely the process from $h_0$ to $h_1$ and back to $\hat{h_0}$, still incurs significant information loss. This aligns with the authors' own admission in the limitations section that the reconstruction loss remains notable. Consequently, I find the validation experiments for BAE insufficient. Although the synthetic dataset confirms the performance of entropy estimation, real-world datasets are unlikely to possess such a clearly defined entropy and composition. Given that entropy is difficult to estimate on real data, I suggest justifying the binarization through alternative methods. For example, the authors could analyze the performance improvements as the mapping granularity is refined—replacing binarization with 4-bit, 8-bit, or 16-bit quantization—to progressively estimate the loss attributable to binarization.

2. More granular evidence comparing the features extracted by BAE and SAE should be provided. For example, which features share similar semantics, or exhibit containment relationships (i.e., one feature being a subset of another)? Which "dead features" from SAE  were not extracted by BAE, or were subsumed into other features? I find the macro-level comparisons in the appendix (e.g., Appendix D) insufficient for what readers are most interested in. Although the authors claim these low-score features are "complex or vague", this merely introduces a new question: are these features genuinely existing, complex phenomena within the LLM, or are they methodological artifacts forcibly produced by BAE during training?

3. The paper's structure could be further improevd. For example, I would recommend moving Appendix D into the main body, whereas some of the "findings" currently in the main text are of less interest. After all, the most significant issue with this paper is the lack of justification.

**Questions:**

1. Could the notations $h_0$ and $h_1$ be changed to something more suitable? The current subscripts imply that $h_1$ is the next layer after $h_0$.

2. I find it difficult to understand BAE's handling of discrete dictionary indices for continuous concepts. For example, if a feature in the dictionary (i.e., in $W_{out}$) represents "emotion," its index activation should correspond to the intensity of that emotion. If this index is binarized, does this imply that only two states exist: "emotion present" and "emotion absent"? Does this not discard too much information?

3. Is it feasible to replace the binarization process with quantization at different bit levels (e.g., 4-bit, 8-bit, 16-bit)? What changes would this introduce?

4. The paper observes that ICL appears as "information reduction". This entropy is calculated on the hidden state of the last token (the ":"). Is this entropy reduction because the model is discarding task-irrelevant information (e.g., noise in the query), or because it is converging all possible answers onto a more deterministic (low-entropy) representation? These two mechanisms are different.

---

### Official Review · Reviewer_Pnrb · 2025-10-26

**Soundness:** 2
**Presentation:** 2
**Contribution:** 2
**Rating:** 4
**Confidence:** 4

**Summary:**

The paper introduces the Binary Autoencoder (BAE) , a novel approach designed to overcome the limitations of traditional Sparse Autoencoders (SAE) in mechanistic interpretability, specifically addressing the issue of dense or simultaneously active features. BAE achieves global feature sparsity and independence by constraining the minimal entropy of binary hidden activations ($h_1 \in \{0,1\}^d$) across training minibatches. This binarization enables efficient and accurate entropy estimation, which is utilized to quantitatively analyze LLM dynamics, such as layer information bandwidth and interpreting In-context Learning (ICL) as information reduction. Empirically, BAE successfully avoids dense features and extracts the largest number of sparse and interpretable features among baselines, evaluated via an adapted Common Semantics-based (ComSem) interpretation method.

**Strengths:**

1. Superior Global Sparsity and Feature Extraction: BAE enforces a global sparsity guarantee by applying a minimal entropy objective on entire minibatches of binary activations. This mechanism effectively mitigates the "dense feature" issue inherent in local, sample-wise regularization of traditional SAEs , leading to the extraction of the largest quantity of atomic and interpretable features.
2. Novel Tool for Information-Theoretic Analysis: By binarizing hidden activations, BAE provides an efficient and reliable method for estimating the entropy of hidden state sets. This capability is leveraged to introduce a new information-theoretic perspective on LLM operations, revealing insights into layer bandwidth limitations and hypothesizing that In-context Learning functions as information reduction.
3. Enhanced Robustness in Feature Interpretation: The authors refine the feature interpretation process with the Common Semantics-based (ComSem) method , which avoids relying on LLMs' unreliable handling of numerical data or abstract output simulation. This revision improves the robustness and credibility of automatically evaluating the interpretability score of extracted features.

**Weaknesses:**

1. The justification for binarization is inadequate, and the results on ICL lack comparative benchmarks. For instance, how does BAE's performance (both with and without demonstrations) compare to the model's original outputs? This could be measured by the difference in KL divergence of the output distributions or other metrics on real data. Given the authors' acknowledgment of significant reconstruction loss, providing more justification for the information loss from binarization is essential; otherwise, binarization itself remains a contentious methodological choice.
2. The ComSem-based evaluation also requires further justification, for instance, by validating it on other mechanistic interpretability tasks such as circuit discovery or causal tracing. Moreover, given that the justification for binarization is already contentious, the ComSem framework feels like a "castle in the air." From my personal perspective, the introduction of ComSem in this paper seems overly bloated.
3. I believe the authors must address a looming concern: To what extent are the "dead features" that SAEs find, but BAEs do not, a result of information loss from binarization, rather than a successful outcome of pursuing true sparsity? This is a difficult point to verify but one that must be confronted. I hope to see thoughtfully designed and sufficient justification experiments from the authors.

**Questions:**

1. What is the theoretical basis for using "Burstiness" to estimate activation magnitude ? What is its precise relationship to information gain?
2. Is the covariance penalty term $D[X]$ necessary? What would be the impact of removing it?
3. Regarding the observation of entropy saturation , is it possible that this merely indicates that, given the current semantic information, a clear dynamic direction has already emerged (e.g., a bottom-up causal emergence in a stochastic process)? Does this necessarily prove that the "implicit context window limit" has been reached?

---

### Official Review · Reviewer_GuWq · 2025-10-30

**Soundness:** 2
**Presentation:** 3
**Contribution:** 2
**Rating:** 4
**Confidence:** 4

**Summary:**

This paper proposes the Binary Autoencoder (BAE), a new autoencoder variant for mechanistic interpretability of Large Language Models (LLMs). Instead of the standard Sparse Autoencoder (SAE) that applies per-instance regularization (e.g., L1 or top-k), BAE introduces a global information-theoretic constraint by minimizing the entropy of minibatch activations.

This approach encourages feature independence and global sparsity, implemented via binarized hidden activations and entropy-based regularization. The paper demonstrates two main applications:
1. Entropy-based analysis — using BAE to measure entropy of hidden states and interpret model behavior (e.g., information bandwidth across layers, and ICL as information reduction).
2. Feature disentanglement — extracting more interpretable and less dense features compared to standard SAEs.

Additionally, the paper proposes ComSem, a modified automatic feature-interpretation framework that avoids numerical token reasoning by relying on semantic similarity judgments from LLMs.

**Strengths:**

1.  Innovative use of information entropy as a training constraint to enforce global sparsity—conceptually elegant and well-justified.
2. Inclusion of inter-feature covariance in the loss function adds an original and reasonable mechanism for promoting feature independence.
3. Insightful findings derived from entropy analysis (e.g., layer-wise entropy dynamics and the interpretation of ICL as information reduction) are thought-provoking and logically consistent.
4. Theoretical clarity—the paper is grounded in information theory and connects entropy minimization with feature disentanglement in a rigorous manner.

**Weaknesses:**

1. Synthetic training data — the benchmark experiments mostly rely on synthetic data. To convincingly demonstrate interpretability, BAE should be trained directly on real LLM hidden states rather than only analyzed on top of them.
2. Weak reconstruction capability — binary representations fundamentally limit expressiveness. As shown in the paper, this leads to relatively high reconstruction loss. This limitation appears to be a trade-off of using entropy as a loss term, but it restricts BAE’s applicability in real-world scenarios where accurate reconstruction matters.
3. Limited novelty of ComSem — while it addresses numerical reasoning issues, its modification from previous evaluation pipelines is minimal. It can be regarded more as an adaptation of existing methods to the binary-activation setting than as a new interpretability framework.

**Questions:**

1. Can the authors train BAE directly on real hidden states (rather than synthetic data) to demonstrate the effectiveness of the entropy regularization?
2. How does the binarization affect the downstream interpretability of reconstructed activations compared to continuous SAEs?
3. Would alternative quantization (e.g., stochastic binarization or ternary encoding) help mitigate reconstruction degradation?

---

### Author Response · Authors · 2025-11-13

We sincerely appreciate the reviewers for their insightful and constructive feedback, from which we have learned a lot. Although we fully agree with the reviewer's comments and guidance, we regret that we are unable to complete the necessary revisions in the discussion period, as we are currently prioritizing other ongoing submissions.

This in no way reflects a lack of respect for the valuable feedback of the reviewers. We sincerely appreciate their thoughtful efforts and will merge their valuable suggestions into our next conference submission.

---

### Meta-Review · Area_Chair_HHPH · 2026-01-08

**Summary:**

This paper develops a new auto-encoder that binarizes latent activations to promote global sparsity and feature independence. The authors are broadly motivated by mechanistic interpretability – e.g., estimating entropy over hidden-state sets and the extraction of interpretable features.  The submission received four reviews with consistent scores. In this case, I  am recommended rejection given the lack of rebuttal and a champion for the work. Although I share some many of the concerns as the reviewers, I believe the overarching problem here stems from the use cases used to motivate the work.  Looking forward, I would recommend thinking of interpretability as a means to an end – i.e., to show that the interpretable features can be used to carry out a useful task in and of itself (e.g., steering or toxicity detection).

**Reviewer Concerns:**

The main concerns are still outstanding.

**Reviewer Scores:**

Given the absence of rebuttal and discussion, I do not expect score changes.

---

### Decision · Program_Chairs · 2026-01-26

Reject